METHODS AND RESOURCES

# Novel kinase regulators of extracellular matrix internalisation identified by high-content screening modulate invasive carcinoma cell migration

**Montserrat Llanses Martinez[1,2], Keqian Nan[1☯], Zhe Bao[1☯], Rachele Bacchetti[1], Shengnan Yuan[1], Joe Tyler[1], Xavier Le Guezennec[2], Frederic A. Bard[2,3], Elena Rainero[1]**\*

**1** School of Biosciences, University of Sheffield, Western Bank, Sheffield, United Kingdom, **2** Institute of Molecular and Cell Biology, Singapore, Singapore, **3** Centre de Recherche en Cancérologie de Marseille, CRCM, Marseille, France

☯ These authors contributed equally to this work.

\* e.rainero@sheffield.ac.uk

**Data Availability Statement:** All relevant data are within the paper and its Supporting Information files.

## Abstract

The interaction between cancer cells and the extracellular matrix (ECM) plays a pivotal role in tumour progression. While the extracellular degradation of ECM proteins has been well characterised, ECM endocytosis and its impact on cancer cell progression, migration, and metastasis is poorly understood. ECM internalisation is increased in invasive breast cancer cells, suggesting it may support invasiveness. However, current high-throughput approaches mainly focus on cells grown on plastic in 2D, making it difficult to apply these to the study of ECM dynamics. Here, we developed a high-content screening assay to study ECM uptake, based on the of use automated ECM coating for the generation of highly homogeneous ECM a pH-sensitive dye to image ECM trafficking in live cells. We identified that mitogen-activated protein kinase (MAPK) family members, MAP3K1 and MAPK11 (p38β), and the protein phosphatase 2 (PP2) subunit PPP2R1A were required for the internalisation of ECM-bound α2β1 integrin. Mechanistically, we show that down-regulation of the sodium/proton exchanger 1 (NHE1), an established macropinocytosis regulator and a target of p38, mediated ECM macropinocytosis. Moreover, disruption of α2 integrin, MAP3K1, MAPK11, PPP2R1A, and NHE1-mediated ECM internalisation significantly impaired cancer cell migration and invasion in 2D and 3D culture systems. Of note, integrin-bound ECM was targeted for lysosomal degradation, which was required for cell migration on cell-derived matrices. Finally, α2β1 integrin and MAP3K1 expression were significantly up-regulated in pancreatic tumours and correlated with poor prognosis in pancreatic cancer patients. Strikingly, MAP3K1, MAPK11, PPP2R1A, and α2 integrin expression were higher in chemotherapy-resistant tumours in breast cancer patients. Our results identified the α2β1 integrin/p38 signalling axis as a novel regulator of ECM endocytosis, which drives invasive migration and tumour progression, demonstrating that our high-content screening approach has the capability of identifying novel regulators of cancer cell invasion.

**Funding:** E.R. is funded by CRUK (C52879/A29144) and this work has also been supported by the Academy of Medical Sciences, Wellcome Trust, Government Department of Business, Energy and Industrial Strategy and British Heart Foundation, Springboard Award (SBF003\1045). M.L.M. is funded by Sheffield/ARAP PhD program. The funders had no role in study design, data collection and analysis, decision to publish, or preparation of the manuscript.

**Abbreviations:** BSA, bovine serum albumin; CDM, cell-derived matrix; DCIS, ductal carcinoma in situ; DMEM, Dulbecco's Modified Eagle's Medium; ECM, extracellular matrix; EGA, European Genome-Phenome Archive; EGFR, epidermal growth factor receptor; ERK, extracellular signal-regulated kinase; FAK, focal adhesion kinase; FBS, fetal bovine serum; GEO, Gene Expression Omnibus; MAPK, mitogen-activated protein kinase; PDAC, pancreatic ductal adenocarcinoma; PP2, protein phosphatase 2; PyMT, polyoma middle T; TCGA, The Cancer Genome Atlas.

## Introduction

The extracellular matrix (ECM) is a noncellular 3D structure that surrounds cells and organs in vivo. Depending on its organisation, it is classified into basement membrane and interstitial matrix. ECM dynamics are tightly regulated in morphogenesis, differentiation, and tissue homeostasis; therefore, dysregulations in ECM remodelling are associated with pathological conditions, including fibrosis and tumour progression. Cells engage with the ECM through the heterodimeric integrin receptors, which are composed of an α and a β subunit and activate diverse pro-survival signal transduction pathways [1,2]. However, the ECM may simultaneously pose a physical constraint for cancer cell invasion and tumour initiation [3,4]. As a result, ECM degradation accompanies tumour progression. Our lab has recently shown that ECM internalisation supports proliferation of breast cancer cells under amino acid starvation by promoting tyrosine metabolism, through a mechanism that requires ECM macropinocytosis, followed by lysosomal degradation [5]. However, the signalling regulators controlling ECM endocytosis are poorly understood.

α2β1 integrin is the major collagen I receptor and is expressed in epithelial cells, platelets, and fibroblasts, among other cell types [6]. α2β1 integrin mediates the reorganisation and contraction of collagen matrices in fibroblasts [7]. In addition, α2β1 integrin has been shown to regulate melanoma and pancreatic carcinoma cell migration [8,9], as well as bone metastatic dissemination of prostate and breast cancer cells [10,11]. However, the mechanisms underlying α2β1 integrin-mediated cancer cell invasive migration have not been elucidated. Integrin/ECM binding triggers the activation of several signalling pathways, including the mitogen-activated protein kinase (MAPK) pathway. In mammalian cells, there are 4 distinct MAPK signalling pathways according to the terminal tier kinase: extracellular signal-regulated kinase (ERK) 1/2, c-Jun N-terminal Kinase (JNK) 1/2/3, p38 and ERK5 [12,13]. p38 MAPK is a family of 4 serine/threonine-specific protein kinases: p38α or MAPK14, p38β or MAPK11, p38γ or MAPK12, and p38δ or MAPK13 [14]. p38α and p38β are ubiquitously expressed and share 75% amino acid sequence homology [14]. p38 MAPK plays an important role in cell proliferation, differentiation, stress responses, autophagy, and cell migration, among other biological processes [14].

A deep comprehension of how cancer cells interact with the ECM is essential to understand tumour growth and metastasis. Here, we showed that ECM internalisation was up-regulated in primary cancer cells extracted from a polyoma middle T (PyMT)-driven mouse model of breast cancer, compared to non-transformed mammary epithelial cells and ECM components were trafficked through EEA1-positive early endosomes and delivered to the lysosomes, where the ECM is targeted for degradation. Through a kinase and phosphatase functional screen, we identified MAP3K1, MAPK11, and PPP2R1A as novel regulators of the macropinocytosis of ECM components, which we defined as being mediated by α2β1 integrin and the Na+/H+ Antiporter, Amiloride-Sensitive (NHE1). Interestingly, invasive breast cancer cells were shown to internalise ECM as they migrated on cell-derived matrices (CDMs) and blocking ECM uptake impaired the invasive migration of breast, pancreatic, and ovarian cancer cells, both in 2D and 3D culture systems. Finally, α2β1 integrin and MAP3K1 were significantly up-regulated in pancreatic tumours compared to healthy tissue and high expression of these genes correlated with reduced survival of pancreatic cancer patients. Remarkably, chemotherapy-resistant breast tumours showed higher mRNA expression levels of MAP3K1, MAPK11, PPP2R1A, and α2 integrin. Altogether, we identified a novel signalling pathway linking ECM macropinocytosis, degradation, and invasive migration in different cancer types, which could potentially be exploited for the generation of novel therapeutic interventions to prevent metastatic dissemination.

## Results

### A kinase and phosphatase screen identifies MAPK signalling cascade as a novel regulator of Matrigel internalisation in breast cancer cells

Polyoma middle T oncogene expression under the mammary epithelial MMTV promoter constitutes a widely used mouse model of breast cancer that recapitulates many aspects of the human disease [15,16]. To assess whether ECM endocytosis is up-regulated in primary breast cancer cells, NMuMG cells, derived from normal mouse mammary glands, and PyMT#1 cells, generated from MMTV-PyMT tumours [15], were seeded on fluorescently labelled Matrigel, a basement-membrane formulation, and CDM, generated by telomerase-immortalised human fibroblasts. It is widely established that, following endocytosis, the ECM is degraded in the lysosomes [17]. Therefore, to prevent lysosomal degradation and discern between changes that could be due to altered ECM degradation rather than endocytosis, cells were treated with a cysteine-cathepsin lysosomal inhibitor, E64d [5,18]. Internalisation of Matrigel was up-regulated in PyMT#1 cells compared to NMuMG cells, both in the presence and the absence of E64d (S1A Fig). Similarly, CDM internalisation was significantly higher in PyMT#1 cells in the presence of E64d (S1B Fig). Conversely, E64d did not increase internalisation of CDM in NMuMG, suggesting that CDM lysosomal degradation is specifically triggered in invasive breast cancer cells (S1B Fig). These data indicate that ECM internalisation and degradation is higher in invasive breast cancer cells, in agreement with our previous results showing that internalisation of collagen I and CDM was increased in the invasive and metastatic MCF10CA1 cells compared to non-transformed MCF10A and non-invasive MCF10-DCIS cells [5].

To determine the trafficking pathway followed by ECM components, we measured Matrigel colocalisation with an early endosomal marker, EEA1, and a late endosomal/lysosomal marker, LAMP2, in MDA-MB-231 triple-negative breast cancer cells. Matrigel colocalisation with EEA1 progressively decreased over time, with the strongest colocalisation observed 3 h after seeding the cells, as indicated by the colocalisation map (S1C Fig). In contrast, Matrigel colocalisation with LAMP2 increased over time and reached a strong colocalisation after 12 h (S1D Fig). These data indicate that Matrigel is first delivered to early endosomes and subsequently to lysosomes for degradation. The endosomal-lysosomal system creates an enclosed environment which is progressively acidified as cargos are transported towards the lysosomes [19]. This difference in pH can be exploited to specifically visualise cargos within the endosomal system, by using pH-sensitive dyes. MDA-MB-231 cells were seeded on collagen I labelled with NHS-fluorescein and a pH-sensitive dye, pHrodo, which fluorescence increases as the endosomal compartment is acidified [20,21]. Using time-lapse microscopy, we showed that as collagen I is internalised, there is a progressive increase in the intensity of pHrodo, while the NHS-fluorescein signal is slightly decreased, consistent with the quenching of the green fluorescence in acidic environments [22] (S1 Video and S1E Fig). These data indicated that following endocytosis, endosomes are rapidly acidified (in about 12 min) and thus pHrodo is a reliable dye for dissecting ECM uptake mechanisms.

To identify the signalling pathways responsible for controlling ECM uptake, we screened an siRNA library targeting 948 protein kinases and phosphatases [23]. MDA-MB-231 cells were knocked down for 72 h, transferred and seeded on pHrodo-labelled Matrigel for 6 h, stained with a nuclear dye and imaged live (Fig 1A). ECM uptake was normalised between the non-targeting (NT) and NT in presence of bafilomycin A1, a V-ATPase inhibitor that prevents lysosomal acidification, as a positive control (Figs 1B, 1C, S2A, and S2B and S1 Table). Moreover, we observed a good correlation between replicates, with $R^2 = 0.6219$ (S2C Fig). Since we previously showed that p21-activated kinase 1 (PAK1) was required for ECM macropinocytosis

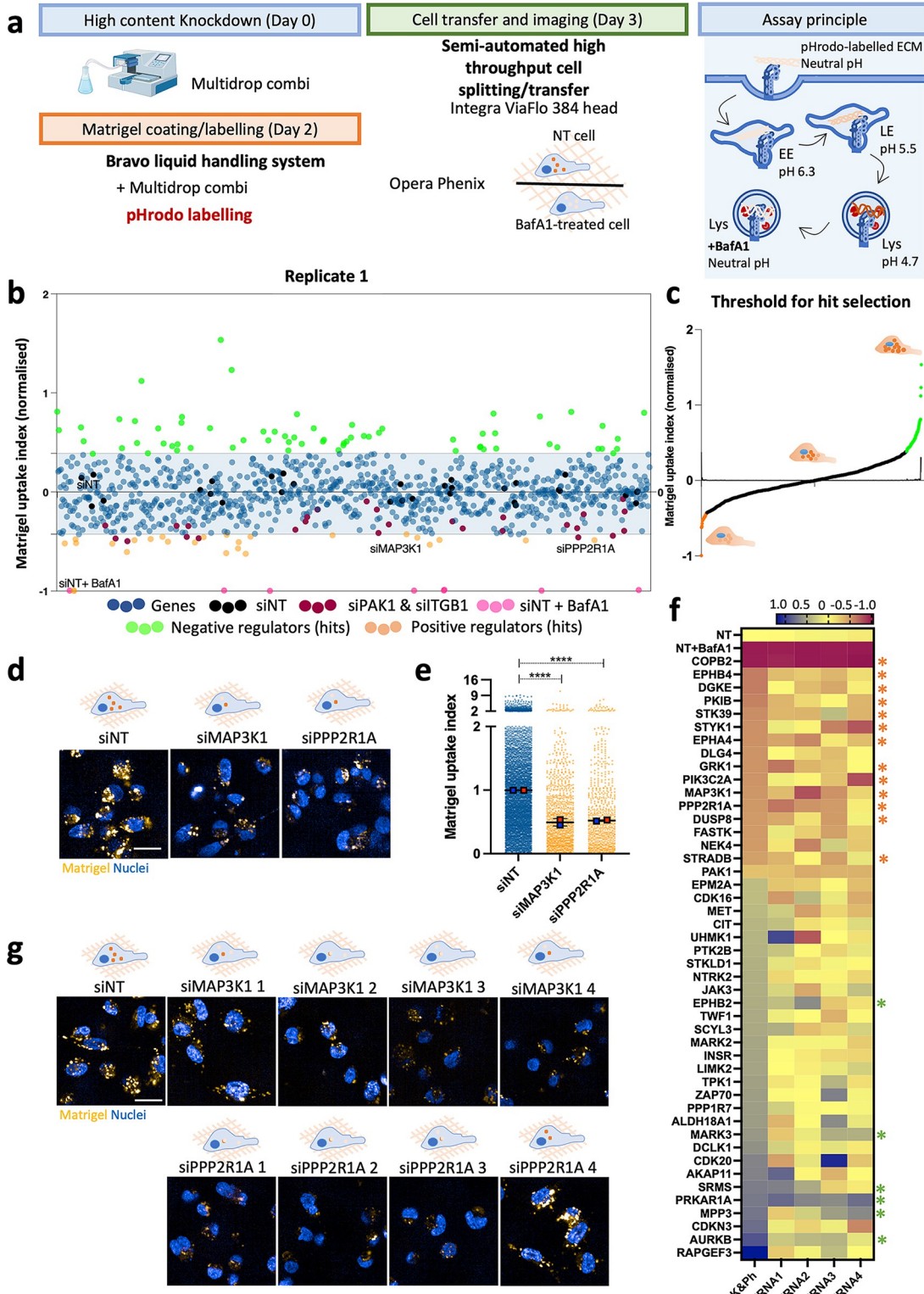

**Fig 1. A kinome and phosphatome screen identified MAP3K1 and PPP2R1A as positive regulators of matrigel internalisation in MDA-MB-231 cells. (a)** Screen schematic representation, 3,000 MDA-MB-231 cells were transfected, transferred into pH-rodo labelled 0.5 mg/ml matrigel, incubated for 6 h and labelled with 1 μg/ml Hoechst for nuclear staining. Cell imaging was carried out with an Opera phenix microscope (40× objective). Columbus software was used for image analysis. **(b)** Normalised cloud plot analysis from replicate 1, positive regulator hits are in orange, negative hits in green. **(c)** First derivative

of the curve for hit threshold for replicate 1. (**d**) Representative images from positive regulators in the screen, MAP3K1 and PPP2R1A. Scale bar, 20 μm. (**e**) Normalised cell data quantification for MAP3K1 and PPP2R1A. Data are presented as the normalised mean ± SD; $N = 3$ independent replicates. ****$p < 0.0001$; Kruskal–Wallis test. (**f**) Heatmap of kinome and phosphatome siRNA deconvolution. $N = 2$ technical replicates, $N = 2$ biological replicates per individual siRNA. Orange stars identified validated positive regulators, green stars validated negative regulators. (**g**) Representative images of the deconvoluted siRNA for MAP3K1 and PPP2R1A. All the raw data associated with this figure are available in S1 Data.

[5], PAK1 was included as an additional positive control. It was reassuring that PAK1 knockdown (**S2G Fig**) consistently reduced Matrigel uptake in the screen (**S2D and S2E Fig**). To control for transfection and silencing efficacy, we knocked down β1-integrin across different plates in the screen. Upon knockdown, β1-integrin intensity was significantly reduced, indicating successful protein knockdown upon siRNA transfection (**S2F Fig**). The Z-factor (Z') provides insight into the quality of a high-throughput assay (where $1.0 > Z \geq 0.5$ indicates an excellent assay and $0.5 > Z > 0$ is considered a marginal assay) [24]. The Z' robust and Z' standard calculated were respectively 0.6 and 0.694, indicating the high quality of the assay (**S2H Fig**).

We utilised Reactome [25] to assess the enrichment for biological pathways in the positive and negative regulators identified by the screen, and enriched pathways included negative regulation of MAPK signalling pathway (*p*-value: 2.68E-05) and MAPK signalling cascades (*p*-value: 3.23E-04; **S2 Table**). MAP3K1 and PPP2R1A were top positive regulators in the screen (**Fig 1D and 1E**). COPB2 knockdown, despite resulting in a strong inhibition in ECM uptake, was not considered further as it was associated with high levels of cell toxicity. MAP3K1 promotes ERK1/2, JNK and p38 activation under distinct stimuli [26,27], while protein phosphatase 2 scaffold subunit α (PPP2R1A) is the regulatory subunit of protein phosphatase 2 A (PP2A) [28]. Consistent with the pathway analysis, we explored additional siRNA targets relevant for MAPK pathway and checked Matrigel uptake upon ERK1/2 (MAPK3 and MAPK1), JNK1/2/3 (MAPK8-10), and p38 (MAPK 11–14) knockdown in the screen. Only MAPK11 (p38β) consistently reduced Matrigel internalisation (**S2I and S2J Fig**), suggesting that MAPK11 signalling is important in this process.

To validate our hits, we performed a secondary screen for 45 common hits obtained between replicates. The pooled siRNAs for each gene were deconvoluted into 4 individual siRNAs (**S3 Table**). We considered it an on target hit if 2 out of 4 siRNAs reduced or increased Matrigel uptake. Based on siPAK1, a 30% reduction or increase compared to NT was used as a cut-off to confirm positive or negative regulators. According to this criterion, 19 genes were validated, among them MAP3K1 and PPP2R1A (**Fig 1F and 1G**). As in the primary screen, knocking down COPB2 strongly reduced both Matrigel uptake index and cell count, indicating cell toxicity. While the individual siRNAs against the positive regulators gave consistent results, this was not the case for most of the negative regulators (on-target hits marked by a green star in **Fig 1F**). Therefore, we focused on the positive regulators of ECM uptake, including MAP3K1, MAPK11, and PPP2R1A.

PP2A is composed of the "C" catalytic subunit, the "B" regulatory subunit, which confers target specificity [29], and the "A" scaffold subunit (**S3A Fig**). The scaffold subunit is encoded by PPP2R1A and PPP2R1B. The former constitutes the PP2A complex in 90% of assemblies [30]. In phosphoproteomic studies, knocking down PPP2R1A has been used to inhibit the whole PP2A phosphatase complex in HeLa cells [31]. At this point, we sought to discern whether other subunits modulated the process in the original screen (**S3B Fig**) and found that only the "C" subunit PPP2CA reduced Matrigel uptake by 25%, maybe due to possible functional redundancies between subunits [32]. Actin polymerisation by the WAVE complex drives macropinocytosis [33] and PPP2R1A was described to form part of the WAVE shell complex, regulating actin polymerisation in a phosphatase activity-independent manner [34].

To discern whether PPP2R1A regulated ECM internalisation through its phosphatase activity or via the regulation of actin polymerisation, we assessed collagen I uptake in the presence of okadaic acid, a PP2A inhibitor [35,36], and found that 50 nM okadaic acid significantly reduced collagen I internalisation after 6 h treatment (S3C Fig), suggesting that the phosphatase function of PPP2R1A regulated ECM uptake. High concentrations of okadaic acid (100–500 nM) have been shown to promote morphological changes, including cell detachment [37]. To rule out that the inhibition of collagen I uptake by okadaic acid was due to cell detachment from the ECM, we performed time-lapse microscopy either in MDA-MB-231 and A2780-Rab25 cells transfected with siPPP2R1A or treated with 50 nM okadaic acid on CDM. Of note, both PPP2R1A knockdown (S3D and S3E Fig) and okadaic acid treatment (S3F and S3G Fig) resulted in cell rounding but not detachment from CDM of either cell line, suggesting that the effect observed was PPP2R1A-specific.

## ECM internalisation is dependent on α2β1 integrin

Our screen identified MAPK signalling as an important regulator of ECM internalisation. Collagens have been linked to the activation of p38 MAPK in platelets [38], while the collagen receptor α2β1 integrin activates p38 MAPK and PP2A in fibroblasts [39,40]. We therefore hypothesised that α2β1 integrin may activate MAPK11 and regulate ECM internalisation. Indeed, down-regulation of β1 integrin significantly reduced Matrigel internalisation in MDA-MB-231 cells (S4A and S4H Fig). To assess whether the ECM was trafficked together with β1 integrin, we assessed β1 integrin localisation in cells seeded on fluorescently labelled Matrigel. Colocalisation analysis revealed a strong overlap between Matrigel and β1 integrin at all the time points measured (S4B Fig). This data supports the hypothesis that β1 integrin not only regulates ECM endocytosis but is trafficked together with the ECM.

In MDA-MB-231 cells, treatment with the α2 integrin pharmacological inhibitor BTT-3033, which disrupts collagen binding, significantly decreased CDM and collagen I endocytosis (Fig 2A and 2B). Consistently, siRNA-mediated down-regulation of α2 integrin (S4G Fig) reduced the internalisation of CDM and collagen I (Fig 2C and 2D). We then wanted to determine whether α2-dependent ECM uptake was shared between different cancer types or was a specific feature associated with invasive breast cancer cells. In ovarian cancer, the overexpression of the small GTPase Rab25 promotes the internalisation of fibronectin-occupied α5β1 integrin to sustain invasive migration [41]; therefore, we measured ECM internalisation in the highly invasive ovarian carcinoma cell line A2780, overexpressing Rab25 (A2780-Rab25). Similar to MDA-MB-231 cells, α2 integrin down-regulation significantly reduced CDM and collagen I uptake in A2780-Rab25 cells (S4C and S4D Fig). To study whether α2 integrin regulated this process in a primary cell line, we treated the polyoma middle T-driven mouse breast cancer cell line YEJ P [42] with BTT-3033. Again, pharmacological inhibition of α2 integrin nearly abolished collagen I internalisation in YEJ P cells (S4E Fig). Similar to breast tumours, pancreatic tumours are surrounded by an extremely dense and fibrotic stroma [43,44] and pancreatic ductal adenocarcinoma (PDAC) cells have been previously shown to be able to internalise collagen I [45]. We assessed collagen I uptake in the metastatic PDAC cell line SW1990 and found that α2 integrin inhibition significantly reduced collagen I internalisation (S4F Fig). Altogether, this data suggests that α2β1 integrin is required for ECM internalisation in different cancer types.

## MAP3K1, MAPK11, and PPP2R1A regulate macropinocytosis of ECM-bound α2β1 integrin

Our data suggested that MAPK signalling regulates ECM endocytosis (S2 Table). To characterise the role of MAPK signalling in this process, we screened 4 inhibitors against p38α/β

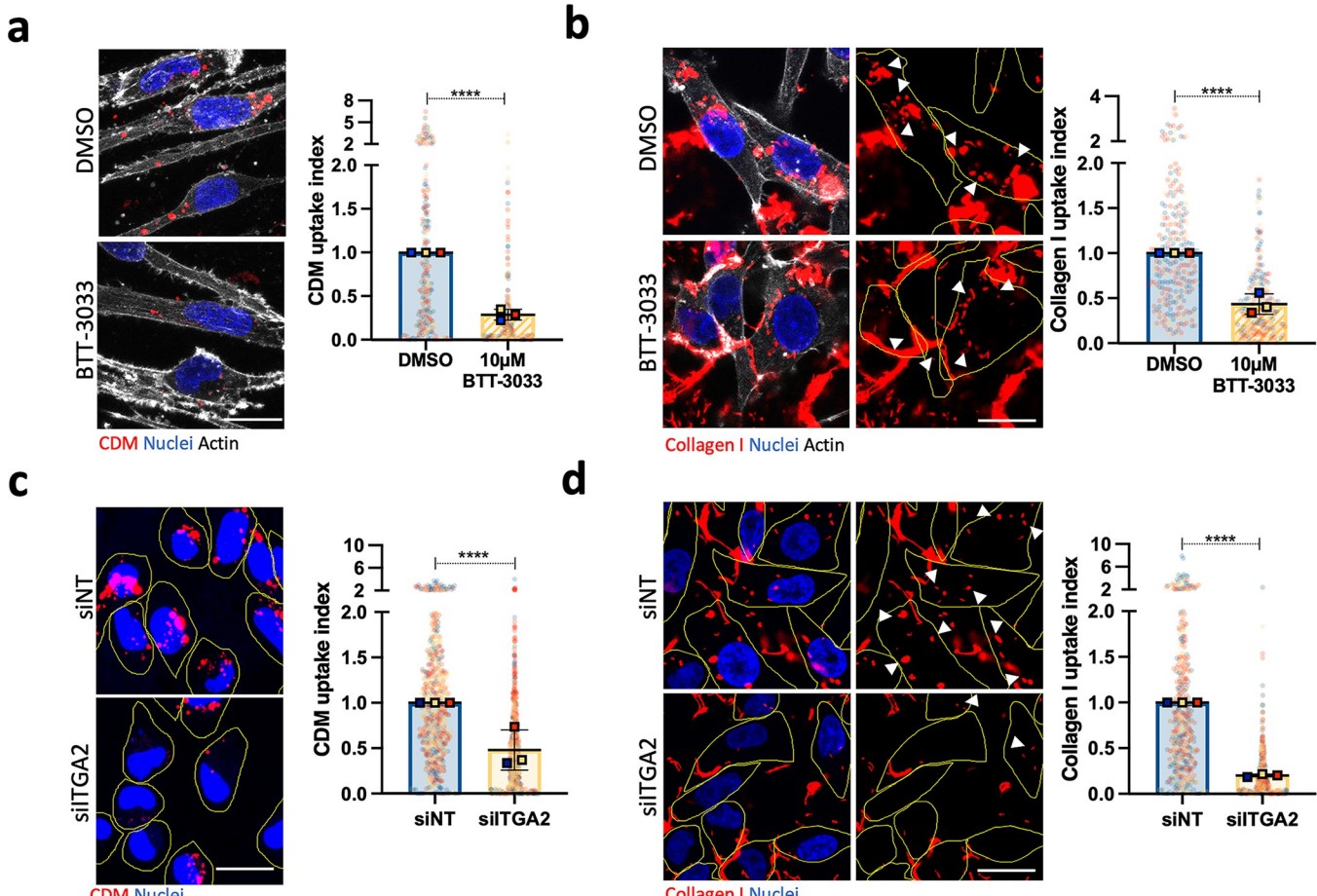

**Fig 2. a2 integrin promoted ECM internalisation in MDA-MB-231 breast cancer cells. (a)** MDA-MB-231 cells were allowed to adhere on biotinylated CDM for 2 h before incubating the cells with 10 μM BTT-3033, an α2 integrin inhibitor, or DMSO for 6 h. Cells were fixed and stained with phalloidin Alexa-Fluor 555, Alexa Fluor 488-streptavidin, and DAPI. Data are presented as the normalised mean ± SD; $N$ = 3 independent replicates. ****$p < 0.0001$; Mann–Whitney test. **(b)** MDA-MB-231 cells were cultured on NHS Alexa Fluor 555-labelled 1 mg/ml collagen I for 2 h before treatment with 10 μM BTT-3033, or DMSO for an additional period of 6 h. Cells were fixed and stained for actin and nuclei. Data are presented as the normalised mean ± SD; $N$ = 3 independent replicates. ****$p < 0.0001$; Mann–Whitney test. **(c)** MDA-MB-231 cells were transfected with an siRNA targeting α2 integrin (siITGA2) or a non-targeting siRNA control (siNT), seeded on pHrodo-labelled CDM for 6 h, stained with 1 μg/ml Hoechst and imaged live. Data are presented as the normalised mean ± SD; $N$ = 3 independent replicates. ****$p < 0.0001$; Mann–Whitney test. **(d)** MDA-MB-231 cells transfected as in (c), seeded on Alexa Fluor 555-labelled 1 mg/ml collagen I for 6 h, fixed and stained for nuclei. Data are presented as the normalised mean ± SD; $N$ = 3 independent replicates. ****$p < 0.0001$; Mann–Whitney test. All the raw data associated with this figure are available in S2 Data. CDM, cell-derived matrix; ECM, extracellular matrix.

(SB202190 and SB203580), ERK1/2 (FR180204), and MEK1/2 (PD98059) on Matrigel, collagen I, and CDM uptake in MDA-MB-231 cells (**S5A Fig**) [46–50]. The strongest uptake impairment was obtained by p38 inhibition in a dose-dependent manner, starting from 10 μM (**S5B Fig**). While 50 μM ERK1/2 inhibition reduced ECM internalisation, inhibition of MEK1/2, an upstream activator of ERK1/2, did not reduce ECM uptake at effective inhibitory concentrations starting from 10 μM (**S5B Fig**). Altogether, this data supported a major role for p38 MAPK in mediating ECM internalisation.

To corroborate these results, highly invasive MDA-MB-231 and A2780-Rab25 cells were seeded on labelled CDM and treated with 10 μM and 50 μM of p38α/β inhibitors. Similarly, p38 inhibition reduced CDM internalisation in a dose-dependent fashion (**Figs 3A, 3B, and S5C**). These experiments were performed with pHrodo-labelled ECM. Under these conditions, a reduction in pHrodo signal could represent either a blockade of endocytosis, a defect

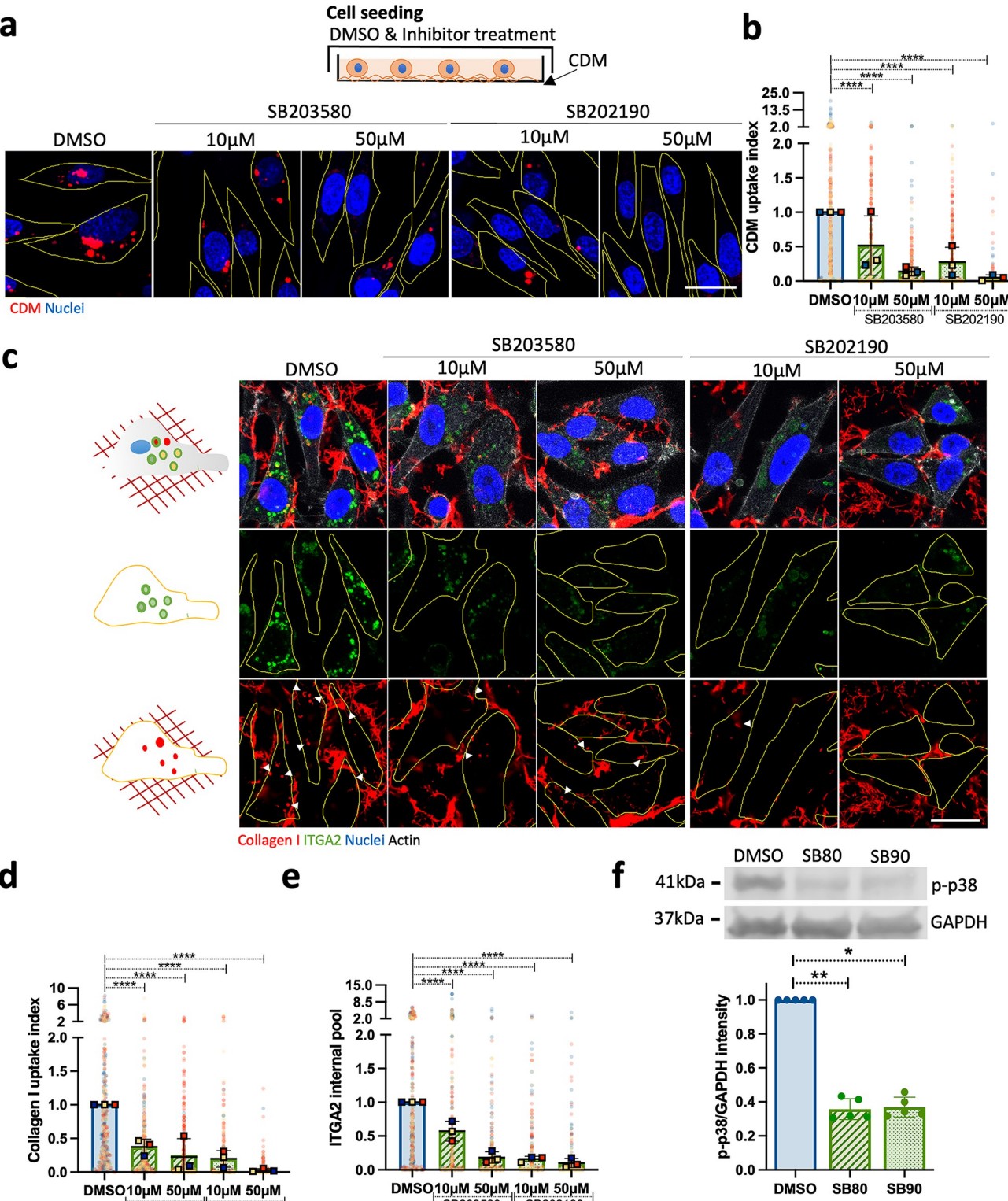

**Fig 3. p38 MAPK regulated the internalisation of collagen I-occupied α2β1 integrin in MDA-MB-231 cells. (a)** MDA-MB-231 cells were serum starved for 16 to 18 h, and $3 \times 10^5$ cells were seeded on pHrodo-labelled CDM for 6 h in the presence of DMSO, 10 μM or 50 μM of SB203580 and SB202190 in 5% FBS and imaged live. Scale bar, 22 μm. **(b)** CDM uptake index was calculated with ImageJ. Values represented are normalised mean + SD from $N = 3$ independent experiments; ****$p < 0.0001$; Kruskal–Wallis test. **(c)** MDA-MB-231 cells were serum starved for 16 to 18 h, and $3 \times 10^5$ cells were seeded on 1 mg/ml collagen I, labelled with NHS-Alexa Fluor 555, treated as in (a), fixed and stained for α2 integrin (ITGA2) and nuclei. Scale bar,

20 µm. **(d, e)** Collagen I uptake index and α2 integrin internal pool were calculated with ImageJ. Values represented are normalised mean + SD from $N = 3$ independent experiments; ****$p < 0.0001$; Kruskal–Wallis test. **(f)** Cells were serum starved for 24 h, preincubated for 1 h with 50 µM SB253080 (SB80), 50 µM SB202190 (SB90), or DMSO, stimulated with 250 mM sorbitol for 15 min and lysed. Phospho-p38 (p-p38) and GAPDH protein levels were measured by western blotting. Data are presented as the normalised mean ± SD; $N = 5$ independent experiments. *$p = 0.0286$; Mann–Whitney test. All the raw data associated with this figure are available in S3 Data. CDM, cell-derived matrix; MAPK, mitogen-activated protein kinase.

in lysosomal targeting of internalised ECM or an impairment of lysosomal acidification. To address this, MDA-MB-231 cells and A2780-Rab25 cells were seeded on collagen I labelled with NHS-Alexa Fluor 555, a non-pH sensitive dye. Consistently, collagen I internalisation was markedly decreased upon p38α/β inhibition (**Figs 3C, 3D, and S5D**), indicating that the observed effect was due to changes in endocytosis and not in endosomal acidification. Similarly, p38 inhibition reduced collagen I internalisation in YEJ P and SW1990 cells (**S5E and S5F Fig**), suggesting that p38 signalling regulates ECM internalisation in breast, ovarian, and PDAC cells. Importantly, these tumours are characterised by a dense and collagen I-rich stroma. Since we showed that β1 integrin trafficked together with Matrigel and α2β1 integrin was required for ECM uptake (**S4B Fig**), we aimed to assess whether p38 inhibition controlled the internalisation of α2β1 integrin, by measuring α2 internal pool. Indeed, p38 inhibition reduced the levels of internalised α2 integrin in MDA-MB-231 cells (**Fig 3C and 3E**). Both inhibitors significantly reduced p38 phosphorylation in MDA-MB-231 cells, indicating inhibition of p38 (**Fig 3F**). In agreement with our previous observations, α2 integrin could be detected in collagen I-positive vesicles (**Fig 3C**), indicating that p38 promotes the internalisation of ECM-bound α2β1 integrin.

To validate the positive regulators identified in our screen, we assessed CDM and collagen I uptake by highly invasive MDA-MB-231 and A2780-Rab25 cells in the presence of siRNA against MAP3K1, MAPK11, and PPP2R1A (**S6E–S6I Fig**) and found that the knockdowns significantly reduced CDM and collagen I uptake in both cell lines (**Figs 4A–4D and S6A–S6D**). Moreover, MAPK11 down-regulation by 3 individual siRNAs sequences significantly impaired collagen I uptake (**S7 Fig**), indicating it is an on-target hit as MAP3K1 and PPP2R1A (**Fig 1F**). In MDA-MB-231 cells, down-regulating MAP3K1, MAPK11, and PPP2R1A also significantly reduced α2 integrin internal pool (**Fig 4C and 4E**). To rule out the possibility that the decrease in internalised α2 integrin was due to changes in its expression, we assessed how MAPK11 and PPP2R1A knockdown affected α2 integrin protein levels by western blotting. Knocking down PPP2R1A resulted in a ~50% decrease in α2 integrin expression (**S6J Fig**), while MAPK11 down-regulation resulted in a small, but not statistically significant increase in α2 integrin levels (**S6K Fig**). These data suggest that changes in the α2 integrin internal pool are likely due to reduced endocytosis upon MAPK11 down-regulation; however, PPP2R1A may potentially regulate both α2 integrin trafficking and expression. Altogether, these data indicate that MAP3K1, MAPK11, and PPP2R1A control ECM uptake by modulating integrin endocytosis. To investigate the fate of internalised ligand-bound α2 integrin, we quantified collagen I and α2 integrin internal pool in the presence of either E64d or Bafilomycin A1, which prevents lysosomal function, and found that the treatments resulted in a significant accumulation of both collagen I and α2 integrin inside the cells (**S8A–S8D Fig**). Consistently, higher α2 integrin protein levels were detected by western blotting upon Bafilomycin A1 treatment (**S8E Fig**), indicating that ligand-bound α2 integrin endocytosis resulted in lysosomal degradation.

We have previously shown that breast cancer cells uptake different ECM components through macropinocytosis [5]. To assess whether p38 was controlling this endocytic process, we measured dextran uptake in MDA-MB-231 cells seeded on collagen I (**S9A Fig**). Interestingly, p38 inhibition significantly reduced dextran internalisation (**S9B Fig**), suggesting that this pathway is a conserved macropinocytic programme. Since integrins cluster in

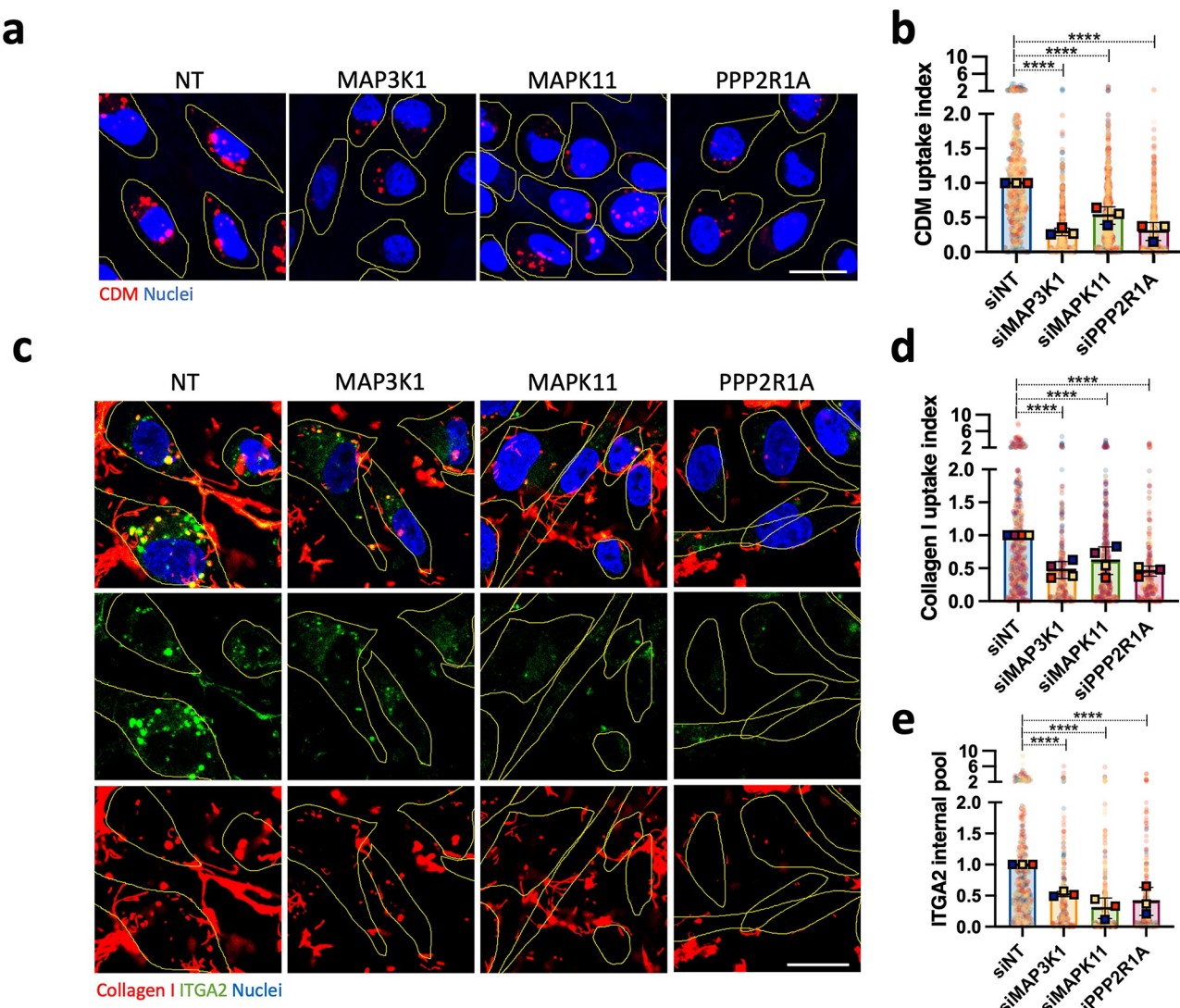

**Fig 4. MAP3K1, MAPK11, and PPP2R1A modulated collagen I-occupied α2β1 integrin uptake. (a)** MDA-MB-231 cells were transfected with an siRNA targeting MAPK3K1 (siMAP3K1), an siRNA targeting MAPK11 (siMAPK11), an siRNA targeting PPP2R1A (siPPP2R1A) or a non-targeting siRNA control (siNT), seeded on pHrodo-labelled CDM for 6 h, stained with 1 μg/ml Hoechst and imaged live. Scale bar, 20 μm. **(b)** CDM uptake index was calculated with ImageJ. Values represented are normalised mean + SD from $N = 3$ independent experiments; ****$p < 0.0001$; Kruskal–Wallis test. **(c)** MDA-MB-231 cells were transfected as in (a), seeded on 1 mg/ml collagen I, labelled with NHS-Alexa Fluor 555, for 6 h, fixed and stained for α2 integrin (ITGA2) and nuclei. Scale bar, 20 μm. **(d, e)** Collagen I uptake index and α2 integrin internal pool were calculated with ImageJ. Values represented are normalised mean + SD from $N = 3$ independent experiments; ****$p < 0.0001$; Kruskal–Wallis test. All the raw data associated with this figure are available in S4 Data. CDM, cell-derived matrix; MAPK, mitogen-activated protein kinase.

macropinocytic cups [33], we reasoned that α2 integrin may be required for macropinocytosis. Indeed, α2 inhibition with either BTT-3033 or siRNA-mediated down-regulation significantly reduced dextran uptake (**S9C and S9D Fig**), indicating that α2β1 integrin is a novel macropinocytosis regulator in invasive breast cancer cells.

## Regulators of ECM internalisation modulate invasive cancer cell migration

Integrin endocytosis is essential for integrin turnover, migration, and invasion [51]. Internalisation of fibronectin-bound α5β1 integrin is required for invasive migration in A2780-Rab25

cells [41], while degradation of the ECM is required in invasive migration [15] to enable cell movement through "ECM barriers" [4]. We observed that MDA-MB-231 cells seeded on 2D collagen I internalised and acidified collagen I while migrating (**Fig 5A and S2 Video**), suggesting that ECM endocytosis might facilitate cell migration. Consistently, breast cancer cells internalised CDM during migration, and this seemed to occur in the perinuclear region in front of the extending protrusion at the leading edge (**Fig 5B and S3 Video**). We therefore reasoned that α2β1 integrin and p38 may facilitate invasive migration by promoting ECM uptake.

To assess this, we treated MDA-MB-231 cells with p38 MAPK inhibitors and found that 10 μM and 50 μM SB203580 and SB202190 significantly reduced the velocity of MDA-MB-231 cells migrating on CDM, while only 50 μM SB203580 and SB202190 impinged on the directionality of cell migration (**Fig 5C**). Correspondingly, down-regulation of MAP3K1, MAPK11, PPP2R1A, and α2 integrin significantly reduced the velocity and directionality of MDA-MB-231 cell migration (**Fig 5D and 5E**). To confirm the relevance of this process to different cancer types, A2780-Rab25 cells were treated with 50 μM SB203580 or siRNAs against MAP3K1, MAPK11, PPP2R1A, and α2 integrin and seeded on CDM. Consistently, A2780-Rab25 cell migration was impaired upon p38 inhibition or siRNA down-regulation of MAP3K1, MAPK11, PPP2R1A, and α2 integrin (**S10A–S10D Fig**). As a WAVE shell complex subunit, PPP2R1A has been shown to modulate migration persistence in the non-transformed mammary cell line MCF10A and MDA-MB-231 cells [34]. To discern if the effect of PPP2R1A on migration required its catalytic activity, cells were treated with okadaic acid. PP2A inhibition significantly reduced the velocity and directionality of cell migration in both MDA-MB-231 and A2780-Rab25 cells, indicating that PP2A phosphatase activity is required for migration (**Figs 5F and S10E**). To study whether regulators of ECM endocytosis similarly affected directional cell migration, we performed a scratch-wound healing assay with a collagen I overlay [52] in SW1990 cells. We found that α2 integrin and p38 inhibition significantly reduced wound healing closure compared to the control (**S10F Fig**). We have previously shown that metabolic adaptations induced by ECM internalisation and lysosomal degradation facilitated invasive cell migration under amino acid starvation [5]. To determine whether ECM-endocytosis-driven cell migration was linked to the ability of the cells to degrade the endocytosed material in the lysosomes or was a mechanism of ECM remodelling to enable cell invasion, we monitored MDA-MB-231 cells migration on CDM in the presence of E64d and Bafilomycin A1. Interestingly, we found that the inhibition of lysosomal function significantly reduced both velocity and directionality of cell migration (**S11 Fig**). Taken together, these data indicate that MAP3K1, MAPK11, PPP2R1A, and α2 integrin, positive regulators of ECM uptake, modulate cell migration, therefore correlating ECM internalisation and lysosomal degradation to cell migration.

p38 has been reported to phosphorylate a variety of downstream targets, including the sodium/proton exchanger 1 (NHE1). Indeed, p38 was shown to directly interact and phosphorylate NHE1 on 1 threonine and 3 serine residues in the C-terminal [53]. Importantly, the phosphorylation was required to trigger the channel activation [54]. As NHE1 is a well-established regulator of macropinocytosis (**S12A Fig**) [55], we hypothesised that p38 might control cancer cell migration through the modulation of NHE1-dependent ECM macropinocytosis. To test this, we assessed the ability of NHE1 knockdown cells (**S12D Fig**) to internalise pHrodo labelled collagen I and found that NHE1 down-regulation significantly impaired collagen I uptake (**S12B Fig**). Moreover, NHE1 knockdown significantly reduced the directionality of MDA-MB-231 cells migrating on cell-derived matrices, without affecting the velocity of cell migration (**S12C Fig**), suggesting that p38 could control ECM uptake through the regulation of NHE1 function.

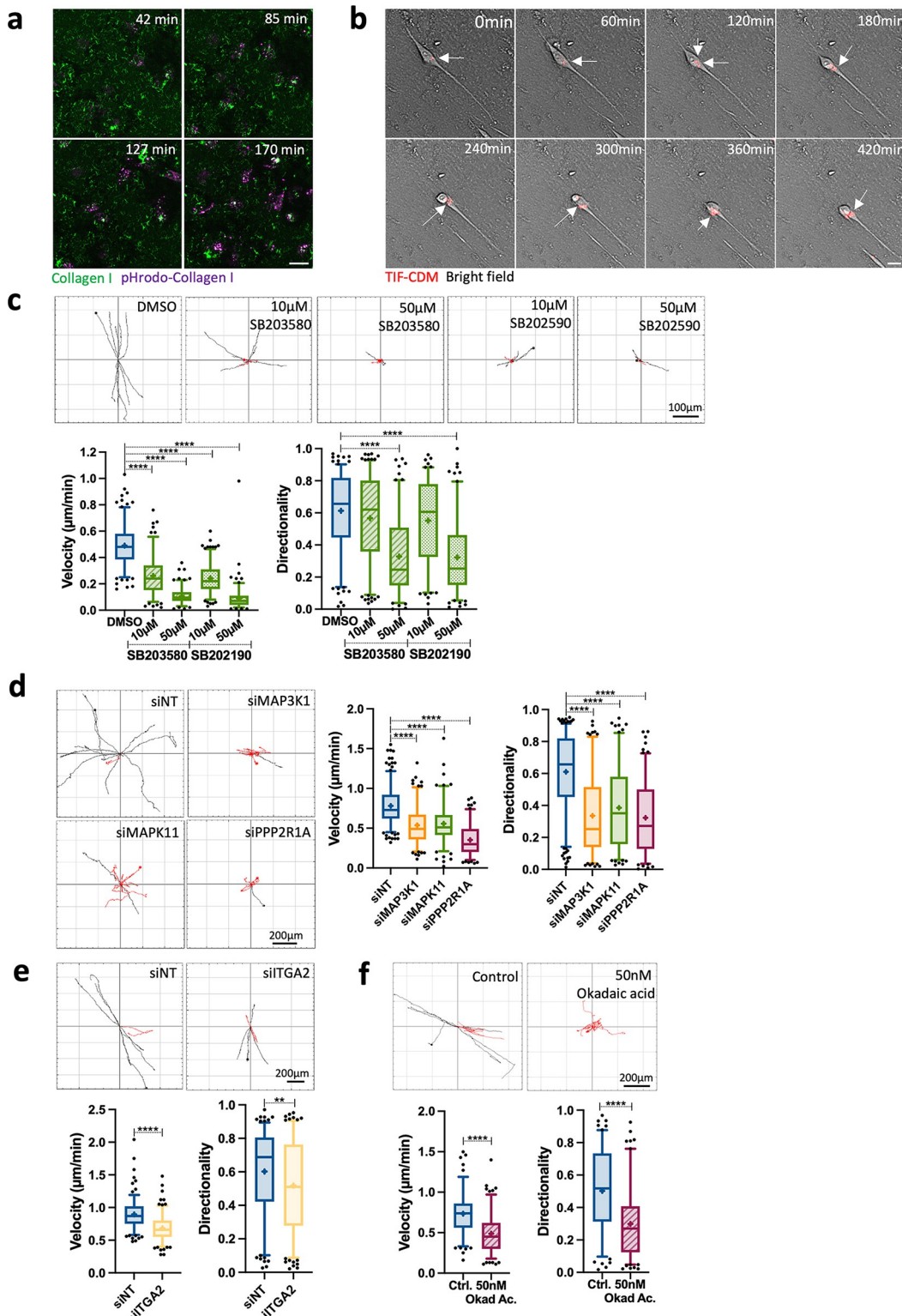

**Fig 5. Regulators of ECM internalisation were required for invasive breast cancer cell migration. (a)** MDA-MB-231 cells were seeded on 1 mg/ml collagen I labelled with pHrodo and NHS-fluorescein for 30 min before live imaging for 5 h. Representative images extracted from **S2** Video are shown. Scale bar, 20 μm. **(b)** MDA-MB-231 cells were seeded in pHrodo-labelled CDM for 6 h and imaged live by time-lapse microscopy. Representative images extracted from **S3** Video are shown. Scale bar, 20 μm. **(c)** MDA-MB-231 cells were seeded on CDM for 6 h in the presence of DMSO, 10 μM and

50 μM SB203580 or SB202190 and imaged live with a 10× Nikon Inverted Ti eclipse with Oko-lab environmental control chamber for at least 7 h. Spider plots show the migration paths of manually tracked cells (directionality >0.5 in black, <0.5 in red). Box and whisker plots represent 5–95 percentile, + represents the mean, dots are <5% and >95%; $N$ = 3 independent experiments. ****$p$ < 0.0001; Kruskal–Wallis test. **(d)** MDA-MB-231 cells were transfected with an siRNA targeting MAP3K1 (siMAP3K1), an siRNA targeting MAPK11 (siMAPK11), an siRNA targeting PPP2R1A (siPPP2R1A) or a non-targeting siRNA control (siNT), seeded on CDM for 6 h and imaged live with a 10× Nikon Inverted Ti eclipse with Oko-lab environmental control chamber for 17 h. Spider plots show the migration paths of manually tracked cells (directionality >0.5 in black, <0.5 in red). Box and whisker plots represent 5–95 percentile, + represents the mean; dots are <5% and >95%; $N$ = 3 independent experiments. ****$p$ < 0.0001; Kruskal–Wallis test. **(e)** MDA-MB-231 cells were transfected with an siRNA targeting α2 integrin (siITGA2) or a non-targeting siRNA control (siNT), seeded on CDM for 4 h and imaged live for 17 h. Spider plots show the migration paths of manually tracked cells (directionality >0.5 in black, <0.5 in red). Box and whisker plots represent 5–95 percentile, + represents the mean, dots are <5% and >95%; $N$ = 3 independent experiments. **$p$ = 0.0032; ****$p$ < 0.0001; Mann–Whitney test. **(f)** MDA-MB-231 cells seeded on CDM for 6 h in the presence of 50 nM Okadaic acid (Okad Ac.) or vehicle control (water) and imaged live for 17 h. Spider plots show the migration paths of manually tracked cells (directionality >0.5 in black, <0.5 in red). Box and whisker plots represent 5–95 percentile, + represents the mean, dots are <5% and >95%; $N$ = 3 independent experiments. ****$p$ < 0.0001; Mann–Whitney test. All the raw data associated with this figure are available in S5 Data. CDM, cell-derived matrix; ECM, extracellular matrix; MAPK, mitogen-activated protein kinase.

Since ECM endocytosis is (1) up-regulated in breast cancer cells; and (2) is required for migration on CDM, we next tested whether regulators of ECM endocytosis were important for 3D invasion (**Fig 6A**). Indeed, down-regulation of MAP3K1 significantly reduced invasion of MDA-MB-231 cells in 3D culture systems (**Fig 6B and 6C**). In agreement with the cell migration data, MDA-MB-231 cells were able to internalise ECM in 3D and MAP3K1 knockdown significantly reduced ECM uptake in these settings (**Fig 6B and 6D**). We also observed differences in the invasive front morphology, where MAP3K1 knockdown resulted in shorter and thicker multicellular strands protruding from the spheroid core (**Fig 6B and 6E**). This observation demonstrates that regulators identified in our 2D screen also play a role in 3D systems, suggesting the screening setup developed in this study has the potential of identifying novel regulators of cancer cell invasion. Similarly to MAP3K1, α2 integrin down-regulation or pharmacological inhibition impaired breast cancer cell invasion in 3D systems (**Fig 6F and 6G**). Altogether, these data demonstrate that MAP3K1 and α2 integrin, regulators of ECM macropinocytosis, are required for MDA-MB-231 cell invasion.

## α2 integrin, β1 integrin, and MAP3K1 are poor prognosis factors for pancreatic carcinoma and are linked to chemoresistance in breast cancer

Since ECM endocytosis is up-regulated in breast cancer cells (**S1A and S1B Fig**), we stained mammary glands of MMTV-PyMT mice extracted at 44 days (normal mammary gland, preceding tumour formation), 73 days (representing ductal carcinoma in situ, DCIS, stage), and 91 days (invasive adenocarcinoma, IDC) for collagen I. Consistent with our in vitro results, more collagen I positive vesicles were present in the IDC tumour, compared to DCIS and normal mammary gland, where a strong collagen I staining can be detected in the stroma (**S13A Fig**). Therefore, we hypothesised that the ECM uptake regulator α2 integrin might be up-regulated during tumour progression in vivo. Indeed, α2 integrin expression was slightly higher in DCIS and IDC tissue sections, compared to the healthy tissues (**S13B and S13C Fig**). In breast cancer, chemotherapy resistance is often accompanied by metastatic dissemination, leading to poor clinical prognosis [56]. To examine whether regulators of ECM internalisation may impact on chemoresistance in breast cancer patients, we looked at the correlation between MAP3K1, MAPK11, PPP2R1A, and α2 integrin expression and chemotherapy response using transcriptomic data of 3,104 breast cancer patients [57]. Interestingly, these genes were highly expressed in chemoresistant (non-responder) tumours (**S13D–S13G Fig**). Indeed, the area under the ROC curve (AUC) for PPP2R1A and α2 integrin was over 0.6 (**S13F and S13G Fig**),

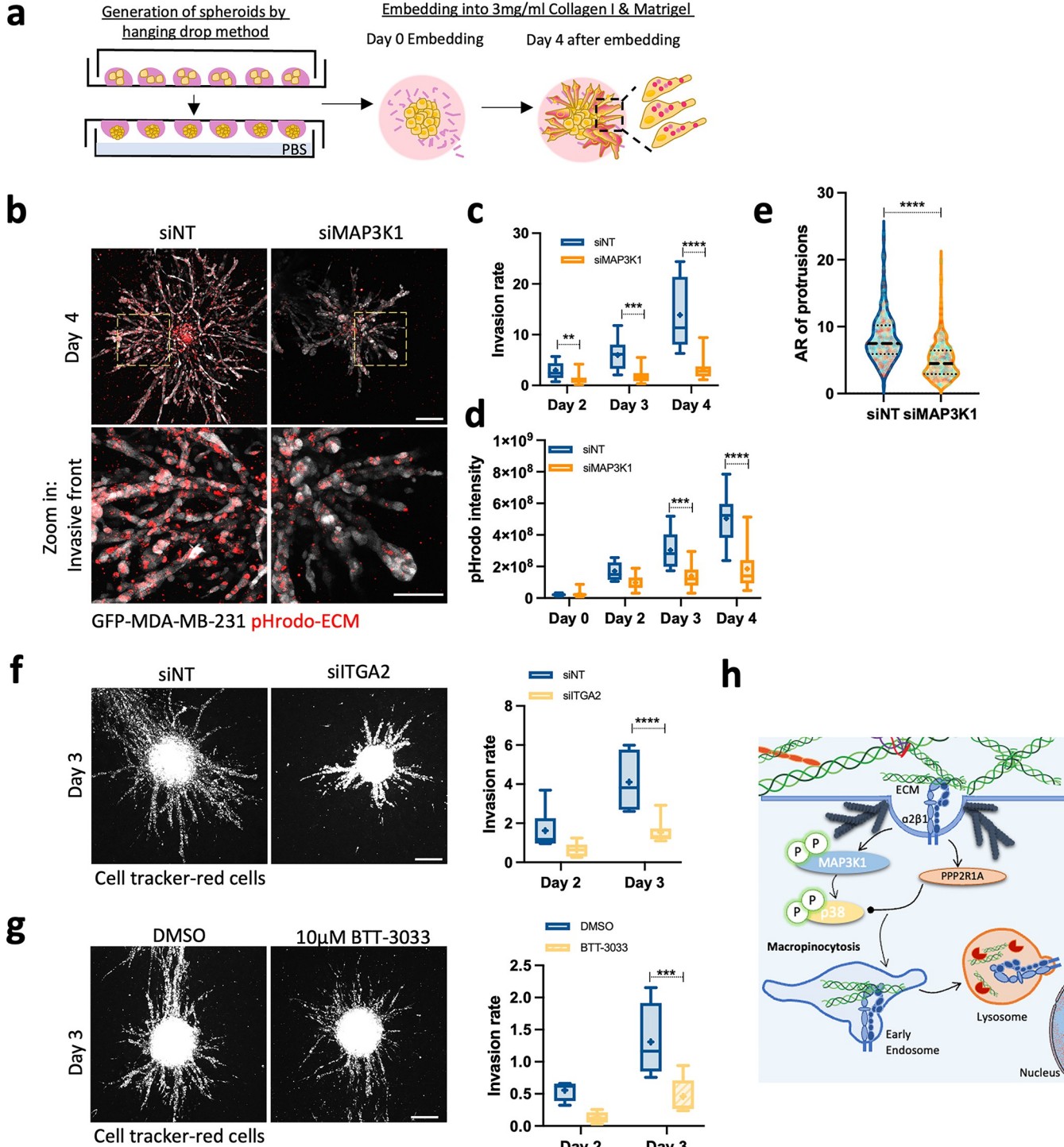

**Fig 6. MAP3K1 and α2 integrin were necessary for MDA-MB-231 cell invasion. (a)** Schematic representation of ECM internalisation in 3D. **(b)** MDA-MB-231 cells were transfected with an siRNA targeting MAP3K1 (siMAP3K1) or a non-targeting siRNA control (siNT) and spheroids embedded in 3D 3 mg/ml pHrodo-labelled collagen I and matrigel (1:1 ratio) mixture for 4 days. Scale bar, 200 μm. For the invasive front: scale bar, 100 μm. **(c)** Invasion rate (spheroid invasion area/spheroid core area). Box and whisker plots represent 5–95 percentile, + represents the mean, $N = 4$ independent experiments. **$p = 0.0060$, ***$p = 0.0003$, ****$p < 0.0001$; 2-way ANOVA, mixed-effects analysis test. **(d)** pHrodo intensity in spheroids. Box and whisker plots represent Min to Max, + represents the mean; $N = 4$ independent experiments. ***$p = 0.0004$, ****$p < 0.0001$; mixed-effects analysis test. **(e)** AR of invasive protrusions. Cell data are presented in violin plots as the median and quartiles; $N = 4$ independent experiments. ****$p < 0.0001$; Mann–Whitney test. **(f)** MDA-MB-231 cells were transfected with an siRNA targeting α2 integrin (siITGA2) or non-targeting siRNA control (siNT) and spheroids were embedded in 2 mg/ml collagen I and

matrigel (1:1 ratio) matrix for 3 days. Scale bar, 200 μm. Box and whisker plots represent Min to Max, + represents the mean; *N* = 2 independent experiments. ****$p < 0.0001$; 2-way ANOVA test. **(g)** MDA-MB-231 cells spheroids were embedded in 2 mg/ml collagen I and matrigel (1:1 ratio) matrix and treated with DMSO or 10 μM BTT-3033. Scale bar, 200 μm. Box and whisker plots represent Min to Max, + represents the mean; *N* = 2 independent experiments. ***$p = 0.0005$; 2-way ANOVA test. **(h)** Working model. All the raw data associated with this figure are available in S6 Data. AR, aspect ratio; ECM, extracellular matrix.

which classified them as a weak biomarker with potential use in prediction of chemotherapy treatment [57]. Given the requirement of α2β1 integrin/p38 MAPK axis in ECM internalisation and invasion in PDAC cells, we looked at the relationship between the expression of α2 integrin, β1 integrin, and MAP3K1 and disease outcomes in PDAC patients. RNA sequencing data showed overexpression of α2 integrin, β1 integrin, and MAP3K1 in pancreatic adenocarcinomas compared to normal pancreas (**Fig 7A, 7D and 7G**). Correspondingly, high expression of these genes correlated with poor overall survival (**Fig 7B, 7E and 7H**) and relapse-free survival (**Fig 7C, 7F and 7I**). These data indicate that α2 integrin and MAP3K1 may contribute to invasion, metastasis, and chemoresistance in vivo.

## Discussion

Dysregulations in endocytic trafficking, such as overexpression of trafficking proteins or macropinocytosis, have been associated with cancer [41,58–60]. We showed that Matrigel and CDM internalisation and degradation are up-regulated in mouse mammary cancer cells, PyMT#1, compared to normal mouse mammary epithelial cells, NMuMG. These data agree with our previous observations using the MCF10 series of non-transformed, DCIS and invasive cells [5]. It is noteworthy that the MCF10 cell lines are epithelial and present cell–cell contact sites, indicating that the changes observed between non-transformed and invasive cells is not due to changes in cell morphology nor cell–cell contact sites. Increased lysosomal degradation of CDM, in contrast to Matrigel, in invasive cancer cells may reflect a characteristic acquired during invasion into the surrounding stroma of primary tumours. In fact, starved mouse mammary epithelial cells were reported to internalise and degrade the underlying basement membrane for survival [61]. Following a canonical endocytic route, Matrigel is first delivered into early endosomes and later to lysosomes. This differs from previous results in which fibronectin-occupied α5β1 integrin was directly delivered into lysosomes [41], suggesting that different ECM components follow distinct trafficking routes. We previously reported that macropinocytosis is the main endocytic route for internalisation of collagen I, Matrigel, and CDM [5]. Macropinosome acidification has been shown to occur within 5 to 10 min after macropinosome formation [62], which is in agreement with our live cell imaging observations, where pHrodo-collagen I fluorescence increased approximately 12 min after internalisation.

Integrin signalling has mainly been associated with Src, focal adhesion kinase (FAK) and epidermal growth factor receptor (EGFR) activation, resulting in cell adhesion, proliferation, and migration [63–65]. While integrins have been shown to modulate ECM uptake, the signalling regulators downstream of integrin engagement that promote ECM internalisation have remained mainly unexplored. We designed a semi-automated high-content screen and found that MAPK signalling, and in particular p38β, is a key regulator of this process. The role of p38 in tumour development remains controversial. Owing to the high sequence homology between p38α and p38β, few studies have focused on deciphering the role of p38β in tumour progression. Nonetheless, recent evidence proposed that p38β may have distinct and nonredundant functions [66]. The p38 inhibitors used in this study target both p38α (MAPK14) and β (MAPK11) [67], we thus cannot discard that under certain circumstances MAPK14 may regulate this process. We previously showed that PAK1 modulates macropinocytosis of Matrigel,

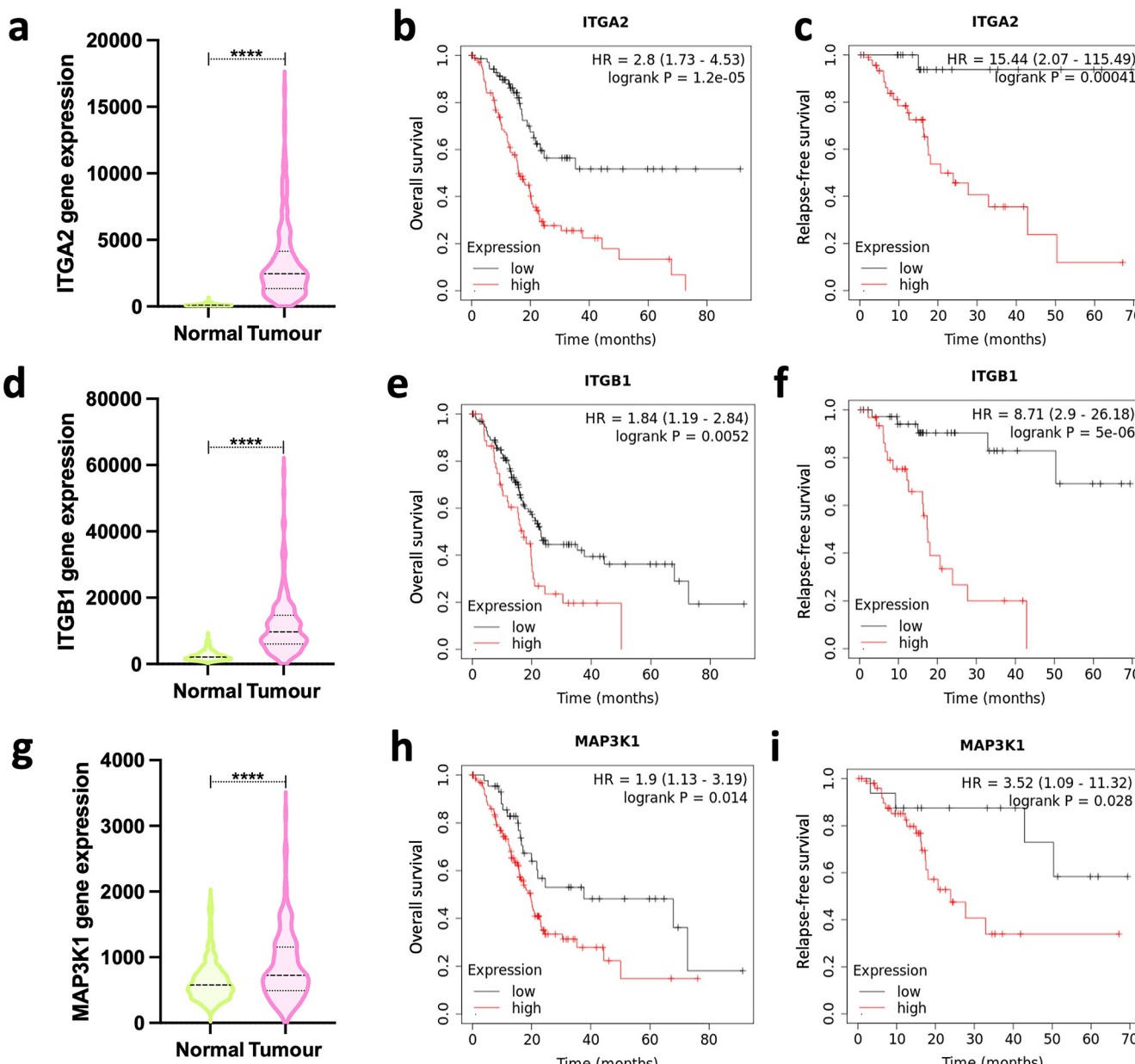

**Fig 7. α2, β1 integrin, and MAP3K1 expression correlated with poor prognosis of pancreatic ductal adenocarcinoma. (a, d, g)** RNA sequencing data from PDAC tumours ($N = 177$) and normal pancreatic tissue ($N = 252$) for α2 integrin (ITGA2, a), β1 integrin (ITGB1, d), and MAP3K1 (g). ****$p < 0.0001$; Mann–Whitney test. **(b, e, h)** Overall survival of patients with PDAC with high (red) or low (black) α2 integrin (ITGA2, b), β1 integrin (ITGB1, e) and MAP3K1 (h) expression. **(c, f, i)** Relapse-free survival of patients with PDAC with high (red) or low (black) α2 integrin (ITGA2, c), β1 integrin (ITGB1, f), and MAP3K1 (i) expression. The data underlying the graphs shown in Fig 7B, 7C, 7E, 7F, and 7H were directly generated using KMplot.com (https://kmplot.com/analysis/index.php?p=service&cancer=pancancer_rnaseq). The rest of the raw data associated with this figure are available in S7 Data. MAPK, mitogen-activated protein kinase; PDAC, pancreatic ductal adenocarcinoma.

collagen I, normal fibroblast- and cancer-associated fibroblast-CDM [5]. PAK1 is recruited to p38 kinase complex in a phosphorylation-dependent manner [68]. Interestingly, PAK1 has been shown to phosphorylate MAP3K1 on serine 67, which inhibits its binding to JNK kinases [69], while MAP3K1 has been described as an upstream regulator of MAPK11 and 14 [70–73]. 3D collagen I has been shown to lead to p38 activation downstream of α2β1 integrin in

mesenchymal cells, such as fibroblasts, in a Cdc42-dependent manner [39]. As PAK1 can be activated by Cdc42, this raises the intriguing hypothesis that PAK1 might promote MAP3K1-dependent p38 activation, leading to ECM uptake, downstream of collagen I binding to α2β1 integrin. Indeed, α2 integrin inhibition or down-regulation significantly decreased the uptake of collagen I-rich matrices in MDA-MB–231, A2780-Rab25, YEJ P, and SW1990 cells. It is important to note that α2 integrin down-regulation had a stronger effect on collagen I than CDM internalisation in A2780-Rab25 cells. This is likely due to the abundance of fibronectin in the CDM, which is internalised in an α5β1-dependent manner in A2780-Rab25 cells [41], while fibronectin uptake in MDA-MB-231 cells is very low. This may be due to Rab25 not being overexpressed in MDA-MB-231 cells [74]. These results agree with previous studies showing that α2β1 integrin regulates collagen I remodelling through its phagocytosis in fibroblasts [75]. In addition, we have shown that α2β1 integrin traffics together with internalised collagen I and the down-regulation of MAP3K1, MAPK11, or PPP2R1A significantly reduced ECM uptake and the internal pool of α2 integrin. In addition to ECM uptake, we showed that α2 integrin and p38 were also required for soluble dextran internalisation. For these experiments, cells were seeded on low collagen I concentrations, which resulted in minimal collagen I uptake. It is therefore unlikely that the reduced dextran uptake is due to dextran mostly binding to collagen I and being internalised together with it. Similarly, NHE1, a well-known regulator of macropinocytosis, is required for ECM internalisation. In line with these results, we have previously shown that 5-(N-Ethyl-N-Isopropyl)amiloride (EIPA), a specific NHE1 inhibitor [76], modulates uptake of matrigel, collagen I and CDMs generated by normal and cancer-associated fibroblasts [5]. Thus, we hypothesise that activation of p38 by α2β1 integrin promotes macropinocytosis of both soluble components, such as dextran, and fibrillar ECM. In fact, p38 inhibition was reported to prevent dextran uptake in dendritic cells [77], while α5β1 integrins [33] and α6β1 [78] have been shown to localise to macropinocytic cups and macropinosomes, respectively.

Protein phosphatases have been commonly studied in the context of negative regulation of MAPK signalling pathways; however, our results may indicate that protein kinases (MAP3K1 and MAPK11) may cooperate with PPP2R1A to promote ECM internalisation. Interestingly, the phosphatase activity of PP2A has been shown to dephosphorylate p38 after its activation downstream of collagen I signalling in platelets [38]. Moreover, PP2A is activated downstream of α2β1 integrin in fibroblasts [40]. PP2A is found in integrin adhesion complexes [79], where it promotes FA maturation and cell migration of the fibrosarcoma cell line HT1080 [80]. We propose that MAP3K1 activation downstream of α2β1 integrin promotes MAPK11 activation, with subsequent PP2A activation to regulate the spatiotemporal activation of MAPK11 (**Fig 6H**). Interestingly, MAPK11/14 activation by extracellular stimuli inhibits endocytic recycling and promotes lysosomal degradation of endocytosed receptors [81], suggesting that activation of MAPK11/14 may regulate ECM trafficking at different stages. Future studies will focus on the mechanism regulating the trafficking of ECM to the lysosomes.

Integrin internalisation and ECM degradation have been associated with invasive migration and metastatic dissemination. Live cell imaging showed that MDA-MB-231 cells internalise CDM while migrating. Interestingly, pharmacological inhibition or siRNA-mediated down-regulation of ECM uptake regulators reduced the velocity and directionality of migrating cells. Of note, 50 μM SB202190 and SB203580 affected both the velocity and directionality of migrating MDA-MB-231 cells, in agreement with MAPK11 knockdown. Nonetheless, 10 μM SB202190 and SB203580 did not affect the directionality of these cells. This could be in part explained by the fact that the lower p38 inhibitor concentrations had a moderate effect on CDM uptake compared to the higher concentrations. In addition, the IC50 of the compounds

for MAPK11 is higher than MAPK14 [82]; therefore, a higher concentration might be required to fully inhibit MAPK11, which appears to be the major regulator of ECM uptake.

The role of macropinocytosis in modulating invasive migration is controversial, as while α5β1 macropinocytosis decreases invasion in the Ewing sarcoma A-673 cell line [33], macropinocytosis of α6β1 promotes glioblastoma cell invasion [78]. Interestingly, PPP2R1A is required for embryonic patterning, primitive streak formation, gastrulation and mesoderm formation [83], which consist of migration processes [84]. This suggests the intriguing possibility that during embryonic development migrating cells remodel the surrounding ECM via macropinocytosis. Indeed, the syncytialisation of placental trophoblasts is accompanied by a strong activation of macropinocytosis [85]. Our results suggest that PP2A activity is required for cell migration of MDA-MB-231 and A2780-Rab25 cells. Indeed, PP2A activity is up-regulated in the osteosarcoma cells LM8, MG63 and SaOS cells [86] and PP2A regulates migration, proliferation, and metastasis of osteosarcoma cells [86]. Nevertheless, PP2A was shown to inhibit cervical cancer cell migration by dephosphorylating JNK, p38, and ERK [87]. This suggests that the role of PP2A in cell migration may be cancer specific. In addition, MAP3K1 down-regulation resulted in changes in the multicellular protrusions projecting from 3D spheroids, suggesting either a possible change in the mode of invasion or reduced invasion with limited effects on cell proliferation, resulting in an accumulation of cells in invasive protrusions.

Immunohistological analysis revealed increased collagen I vesicular staining in DCIS and IDC mouse mammary tumours, which may indicate increased ECM internalisation in vivo. However, we cannot rule out the possibility that the intracellular collagen I dotted structures may be due to increased collagen I expression/secretion, associated with acquisition of mesenchymal traits [88]. We observed that α2 integrin expression increased in breast and pancreatic tumours and correlated with poor survival. In agreement with our results, patient-derived xenografts from breast cancer bone metastasis showed increased α2β1 integrin expression when undergoing epithelial-to-mesenchymal transition with progressive passages through mice [89]. Similarly, α2 integrin expression is induced in ovarian cancer cells that metastasized to the omentum [90]. The role of α2 integrin in pancreatic cancer remains controversial. While α2 integrin expression correlates with poor overall survival and resistance to gemcitabine in high-stiffness matrices in pancreatic cancer [91], poorly differentiated tumours were reported to express low levels of α2 integrin, while well-differentiated ones showed high levels of α2 integrin [92]. Consistent with our data showing a correlation between MAP3K1 levels and poor prognosis in PDAC patients, MAP3K1 expression correlates with progression and poor prognosis of hormone-receptor-positive, HER2-negative early-stage breast cancer patients [93]. Single-cell transcriptomes from matched primary tumours and metastasis from patient-derived xenograft models of breast cancer showed that metastasis displayed increased stress response signalling during metastatic progression [94]. Interestingly, p38β (MAPK11) activation has been associated with stress signalling [95], suggesting the intriguing hypothesis that ECM uptake might be induced by the stress response during cancer dissemination. Indeed, MAPK11 promotes metastatic dissemination to the bone, where it promotes osteolytic bone destruction [96]. Similarly, α2 integrin expression promotes bone metastasis of the prostate cancer cell line LNCaP PCa [97]. In addition, integrins and stress responses have been linked to therapy resistance [98]. Indeed, MAPK14 is up-regulated in BRAF-inhibition-resistant melanoma in mouse models [98,99], while MAPK11 abrogation promotes radiosensitivity in A549, MCF7, and HCT-116 cells [100]. Consistently, we showed that MAP3K1, MAPK11, PPP2R1A, and α2 integrin expression is higher in chemoresistant breast cancer patients. Dual inhibition of MAPK11 and MAPK14 is currently being tested as a monotherapy or in combination with other agents, such as gemcitabine and carboplatin, for the treatment of glioblastoma, ovarian, and metastatic breast cancer. Interestingly, the combination therapy improved

progression-free survival in epithelial ovarian cancer [101]. A recent preprint has shown that in patients resistant to PD1/L1 therapy (nivolumab), combination treatment with p38 inhibition (pexmetinib) increased T cell infiltration and clinical response [102]. Similarly, α2 integrin inhibitors are being studied in clinical trials for solid tumours, either alone [103] or in combination with chemotherapy [104]. This raises the intriguing hypothesis that MAPK11/14 activation in therapy-resistant cancers could contribute to metastatic dissemination by promoting ECM-bound α2β1 integrin internalisation. Further work will look into this.

Altogether, we showed that ECM internalisation is up-regulated in breast cancer cells and ECM macropinocytosis facilitates invasive migration of breast, ovarian, and pancreatic cancer cells, through the activation of p38 signalling. High expression of the regulators of ECM internalisation correlates with poor prognosis of pancreatic cancer patients and is associated with chemoresistance in breast cancer, therefore targeting ECM macropinocytosis could open new avenues for the development of novel strategies to prevent cancer cell invasion and dissemination, which ultimately constitutes the primary cause of cancer death [105].

## Methods

### Antibodies and reagents

Primary antibodies Alexa-Fluor 488 Anti-human CD29 (ITGB1, Clone TS2/16; 303016), FITC anti-human CD49b (ITGA2; Clone P1E6-C5; 359306), Alexa-Fluor 488 anti-mouse CD49b (Clone HMα2; 103510), and mouse anti-human LAMP2 (354303) from BioLegend, primary antibodies mouse anti-human CD49b (611017) mouse anti-EEA1 (610457) from BD Bioscience. Primary antibodies anti-human GAPDH (SC-47724) and NHE1 (sc-136239) from Santa Cruz Biotechnology. Primary antibody against p38 MAPK from Cell signalling (9212S). Primary antibody against phospho-p38 MAPK (Rabbit PolyAb Thr180/Tyr182) from Proteintech (28796-1-AP). Primary antibody against collagen I (NB600-408) from Novus. Secondary antibodies Alexa-Fluor 488 Donkey Anti-mouse were from Fisher Scientific (A-21202), IRDye 800CW and IRDye 680CW were from LI-COR. Alexa Fluor 647 Phalloidin, NHS-Fluorescein, NHS-Alexa Fluor 555, pHrodo iFL STP ester red, and Hoechst-33342 were from Invitrogen. Collagen I and Matrigel were from Corning. DMEM, RPMI, OptiMEM, Trypsin-EDTA, TrypLE, and FBS were from Gibco. E64d (Aloxistatin) was from AdooQ Bioscience. Bafilomcyin A1, FR180204, PD98059, SB202190, and SB203580 from MedChem express. SB203580 from Startech. SB202190 from APExBIO. Okadiac acid from Merk. BTT-3033 from Tocris. Bafilomycin A1 from VWR. siRNA from Dharmacon Horizon discoveries. Vectashield antifade mounting medium from VECTOR laboratories.

### Cell culture

MDA-MB-231 cells, MDA-MB-231 cells overexpressing PPP2R1A-GFP, MDA-MB-231 cells expressing GFP, Telomerase immortalised normal fibroblasts (TIF) and SW1990 cells were cultured in high glucose Dulbecco's Modified Eagle's Medium (DMEM) supplemented with 10% fetal bovine serum (FBS). MDA-MB-231 cells overexpressing PPP2R1A were a gift from Professor Alexis Gautreau (École Polytechnique, Paris, France). A2780 overexpressing Rab25, A2780-Rab25, were maintained in Roswell Park Memorial Institute (RPMI) medium supplemented with 10% FBS. NMuMG cells were cultured in DMEM supplemented with 10% FBS and 10 μg/ml insulin. PyMT#1 and YEJ P cells were cultured in DMEM 20 ng/ml EGF and 10 μg/ml insulin. NMuMG, PyMT#1, and YEJ P cells were a gift from Professor Jim Norman (CRUK Scotland Institute, Glasgow, United Kingdom). Cells were grown at 5% $CO_2$ and 37˚C and passaged every 3 to 4 days.

## Generation of cell-derived matrices (CDMs)

CDMs were generated as previously described [106], either in a 35-mm glass bottom dish, 12-well plate, 8-well chamber, or a 384-well plate. Tissue culture plates were first coated with 0.2% (v/v) gelatin for 1 h at 37˚C. Following that time, plates were washed twice with PBS and crosslinked with 1% (v/v) sterile glutaraldehyde (dissolved in PBS) for 30 min at room temperature. Plates were thereafter washed twice with PBS and the remaining glutaraldehyde was quenched with 1 M sterile glycine for 20 min at room temperature. Subsequently, plates were washed twice with PBS and equilibrated for 30 min in complete medium at 37˚C. Confluent TIFs were seeded onto the gelatin-coated plates (S4 Table). TIFs were incubated at 37˚C in 5% $CO_2$ until being fully confluent. The following day or the day after, the media was changed to complete media supplemented with 50 µg/ml ascorbic acid, the media was refreshed every other day. TIFs were kept secreting CDM for 9 days in a 10 cm dish, 12-well plate and 35 mm glass bottom dish. While only 7 days were required for CDM production in 384-well plates. Following that time, cells were washed once with PBS containing $CaCl_2$ and $MgCl_2$ ($PBS^{++}$). Cells were incubated with the extraction buffer (20 mM $NH_4OH$ and 0.5% triton X-100 in $PBS^{++}$) for 2 to 5 min at room temperature until no visible cells remained. For 384-well plates, cells were extracted twice for 2 min. Extracted CDMs were subsequently washed twice with $PBS^{++}$ and residual DNA was digested with 10 µg/ml DNase I in $PBS^{++}$ at 5% $CO_2$, 37˚C for 1 h; for 384-well plates, DNase incubation was overnight. CDMs were then washed with $PBS^{++}$ and stored at 4˚C in $PBS^{++}$ supplemented with 1% Penicillin/Streptomycin.

## High-throughput ECM internalisation screen

*Matrigel coating and labelling.* Matrigel preparations were handled using high-grade dispensing pipette tips (E1-ClipTip Equaliser pipette). All the reagents were added in a high-throughput fashion by using the multidrop combi at slow or medium speed. The small and large multidrop combi dispenser cassettes, henceforth small or large cassettes, were sterilised with 100 ml of 70% ice-cold ethanol. Cassettes were then rinsed with sterile ice-cold water (Gibco). The large cassette dispensed 50 µl ice-cold PBS in a 384-well plate. PBS plates were kept at 4˚C for 15 min. The small cassette was kept at 4˚C and was set to dispense 2 µl ice-cold Matrigel in the 384-well plates containing PBS. Plates were centrifuged for a few seconds at 500 rpm, kept at 4˚C for 8 min and then polymerised for 2 h and 30 min at room temperature. Bravo liquid handling system (Agilent Technologies, hereafter Bravo) was used to perform Matrigel washes. After polymerisation time was completed, 40 µl PBS was pipetted up and deposited into the waste reservoir. The multidrop combi large cassette was set to dispense 30 µl/well of 20 µg/ml pHrodo in a 384-well plate. Plates were kept in the dark on gentle rocking for 1 h for efficient Matrigel labelling. Following that time, Bravo was used for washing pHrodo off. All washes were performed with PBS containing 1% antibiotic-antimycotic (anti/anti) to avoid contamination. Plates were kept in 50 µl 1% anti/anti in PBS at 37˚C overnight. *Cell detachment.* VIA-FLO-384-well head (Integra) was used for cell transfer. Cell media in transfected plates was automatically pipetted and released in a waste reservoir. Transfected plates were washed with 80 µl PBS once and 20 µl of TrypLE was used to ensure the detachment of cells. Cells were incubated for 5 min at 37˚C. Cells were then vortexed every 1 min thrice. TrypLE Express Enzyme was neutralised with 80 µl 10% FBS DMEM. Cells were pipetted up and down before transferring to ECM-coated 384-well plates. Transfected and transferred cells were incubated for 6 h. After 2 h, cell media was changed to 200 nM Bafilomycin A1 in positive control wells containing NT. After 5 h and 45 min, Hoechst-33342 was spiked into a final concentration of 0.5 to 1 µg/ml. Cells were imaged with 40× water objective Opera Phenix. The screen was performed in duplicates (biological replicates); the same imaging and analysis threshold was

applied for both replicates. Data was normalised between the average of the non-targeting (NT) control and NT in the presence of Bafilomycin A1, as a positive control to assess the magnitude of modulation between technical replicates, as in [107]. To identify the hits, we calculated the first derivative of the curve of normalised values within the population tested, excluding controls. We identified 25 and 37 positive regulators, and 66 and 55 negative regulators for the first and second replicate, respectively. The correlation and reproducibility of the hits were the main criteria for selecting the hits; this approach has been conventionally followed in many studies [23,108–110]. To rule out that changes in ECM uptake were a consequence of changes in apoptosis, survival, and/or proliferation, we assessed the nuclei count and considered a nuclei count lower than 200 as an indication of cell toxicity of the knockdown as in [23].

## ECM internalisation

Collagen I was dissolved in ice-cold PBS to a final concentration of 1 mg/ml, and 100 μl of the solution was used to coat a 35 mm glass-bottom dish with the help of a pipette tip. ECM-coated dishes were incubated at 37°C and 5% $CO_2$ for 1 h for polymerisation. Collagen I dishes were labelled with either 300 μl of 10 μg/ml NHS-fluorescein, 5 μg/ml Alexa Fluor 555 NHS ester, or 20 μg/ml pHrodo iFL red. The dishes were incubated for 1 h at room temperature on gentle rocking. CDMs were labelled with 20 μg/ml pHrodo iFL red for 1 h at 37°C. The labelled ECM dishes were washed twice with PBS prior to cell seeding. To avoid evaporation, PBS was added to each dish to keep in the incubator. Alternatively, CDMs were labelled with 0.13 mg/ml NHS-SS-biotin in PBS$^{++}$ for 30 min with gentle rocking at 4°C, labelled CDMs were washed twice, and cells were seeded on top. Cells were then washed once with ice-cold PBS$^{++}$ and treated with a cell-impermeable reducing agent (15 mg/ml sodium 2-mercaptoethane sulfonate supplemented in 3 mM NaOH for 1 h and 30 min at 4°C). Reduced cysteines were alkylated with 17 mg/ml iodoacetamide for 10 min at 4°C. Dishes were kept on ice, washed once with PBS$^{++}$ and fixed with 4% formaldehyde. Cells were permeabilised and stained with Streptavidin Alexa Fluor 488 (1:1,000) at room temperature for 1 h. For pHrodo-labelled ECM, cells were stained with Hoechst-33342 and imaged live. For p38 MAPK inhibitors, cells were serum starved for 16 to 18 h. Cells were detached using TrypLE and neutralised in serum-free media. Preliminary screening of MAPK inhibitors was performed in serum-free conditions to assess ECM-specific contribution and avoid MAPK activation by growth factors found in serum [39]; other assays were performed in 5% FBS. For uptake, cells were incubated for 6 h, except for YEJ P cells, which were kept for 16 h. For BTT-3033 experiments, cells were allowed to adhere for 2 h before adding 5 μM or 10 μM BTT-3033 depending on the cell line. MDA-MB-231 cells were cultured for an additional period of 6 h in the presence of BTT-3033 before fixation and immunofluorescence staining. SW1990 cells were cultured for 4 additional hours before nuclei labelling and live imaging. For knockdown experiments, cells were incubated for a total of 6 h on the different types of ECM. Cells were imaged using a 60× Nikon A1 confocal microscope. For these experiments, cells were stained for a membrane protein, which is not shown in the images for better visualisation of the uptake. For live imaging uptake, the outline of the cells was visible, therefore being used to calculate the cell area. Confocal experiments were analysed manually. For time-lapse uptake, MDA-MB-231 cells were seeded on collagen I matrices labelled with NHS-fluorescein and pHrodo iFL red for 30 min and imaged every minute with a 63× oil objective Zeiss LSM980 Airyscan 2 microscope for a total period of 5 to 6 h.

## Rhodamine-dextran internalisation

MDA-MB-231 cells were seeded at a density of $10^5$ cells/well for 6 h on 0.1 mg/ml collagen I (50 μl/well)-coated 8-well chamber. Then, cells were pretreated with DMSO or 50 μM

SB202190 for 30 min. Cells were later incubated for 1 h with 0.25 mg/ml Rhodamine-dextran in the presence of SB202190 or DMSO. Alternatively, MDA-MB-231 cells were seeded at a density of $3 \times 10^5$ cells/dish on 35 mm glass-bottom dishes coated with 0.1 mg/ml collagen I for 5 h in the presence of BTT-3033 or DMSO. Cells were later incubated for 1 h with 0.25 mg/ml Rhodamine-dextran in the presence of the inhibitors or the vehicle control. Cells were then fixed with 4% paraformaldehyde and stained for human β1 integrin (1:400 dilution). Vectashield mounting medium with DAPI allowed the visualisation of the nuclei. Cell imaging was carried out with a 60× objective Nikon A1 confocal microscope. Fiji/Image J [111] was used to calculate Dextran uptake index [112].

## Immunofluorescence

Cells were fixed with 4% (w/v) formaldehyde in PBS for 15 min. Next, cells were permeabilised with 0.25% (v/v) Triton X-100 in PBS for 5 min and washed twice with PBS. For dextran internalisation assays, cells were not permeabilised. For ECM internalisation assays, cells were stained with Phalloidin conjugated with Alexa Fluor 555, diluted 1:500 in PBS; cells were incubated for 10 min at room temperature. For antibody staining, cells were blocked in 1% (w/v) bovine serum albumin (BSA) for 1 h at room temperature. Cells were then incubated with the respective primary antibodies for 1 h at room temperature. Anti-human ITGB1 antibody conjugated to Alexa Fluor 488 (1:400 in PBS) was used for the colocalisation experiments. For α2 integrin staining, cells were incubated with FITC anti-human CD49b antibody (1:200 dilution) in PBS. For non-fluorescently conjugated antibodies, EEA1 and LAMP2, cells were first incubated with the primary antibody (1:100 in PBS) for 1 h at room temperature. Cells were then washed thrice with PBS and incubated with the secondary antibody Alexa Fluor 488 Donkey Anti-mouse (1:1,000) for 45 min at room temperature. Cells were washed 3 times with PBS and once with ionised water. Vectashield antifade mounting media with DAPI was used for nuclear staining and sample preservation. For colocalisation experiments, the "colocalization colormap" ImageJ plug-in was used [113].

## siRNA transfection

For 384-well plates, 2.5 μl of 500 nM siGENOME siRNA and 2.5 μl Opti-MEM per well were added into CellCarrier Ultra 384-well plates (Perkin Elmer), and 4.95 μl Opti-MEM was incubated with 0.05 μl Dharmafect IV for 5 min, and 5 μl of the Dharmafect IV solution was added into each well. Plates were incubated for 20 min on gentle rocker at RT; $3 \times 10^3$ cells were seeded in 40 μl DMEM containing 10% FBS. The final concentration of the siRNA was 25 nM. Cells were kept at 5% $CO_2$ and 37˚C for 72 h. β1 integrin knockdown efficiency was analysed in Columbus software. For 6-well plates, 10 μl 5 μM siRNA (**S5 Table**) were mixed with 190 μl Opti-MEM into each well of a 6-well plate; 198 μl Opti-MEM and 2 μl Dharmafect I were mixed and incubated for 5 min at RT. A total of 200 μl of the Opti-MEM Dharmafect I mix was added on top of the siRNA and incubated for 20 min on a rocker, and $4 \times 10^5$ cells in 1.6 ml were added into each well. Cells were incubated at 37˚C and 5% $CO_2$ for 72 h. Following this time, cells were used for uptake experiments or analysis of knockdown efficiency by western blot or RT-qPCR. For PPP2R1A knockdown, cells were seeded on a glass bottom dish and knockdown efficiency was analysed in ImageJ. Alternatively, for ITGA2 knockdown migration and invasion assays in MDA-MB-231 cells, cells were seeded in 6-well plates. The next day, 5 μl of Lipofectamine-2000 were mixed in 250 μl of Opti-MEM (Solution A) and 5 μl of 20 μM siRNA were mixed with 250 μl Opti-MEM (Solution B). Both solutions were mixed and incubated for 20 min at room temperature. Plates containing adhered cells were washed once with PBS and 500 μl of Solution A+B was added on top of each well, 500 μl of Optimem was added

and cells were incubated for a period of 4 to 6 h. After this time, the media was aspirated and 2 ml of fresh 10% DMEM was added; alternatively, for 3D spheroid generation, knockdown cells were cultured into methylcellulose drops for 48 h.

## Western blotting

Confluent 6-well plates were harvested with 100 µl of lysis buffer (50 mM Tris-HCl (pH 7) and 1% SDS). For p38 activation/inhibition, cells were serum starved for 24 h, pretreated with SB202190, SB253080, or DMSO for 1 h before being treated with 250 mM sorbitol for 15 min. Cell lysates were collected and transferred into QiaShredder columns (Qiagen), which were spun at 4°C for 10 min at 13,000 rpm. Extracted proteins were mixed in a 4:1 ratio with NuPAGE buffer with a final concentration of 1 mM DTT; 15 µl to 25 µl of extracted proteins in NuPAGE buffer and 0.5 µl protein ladder (BioLabs) were loaded into a Bio-Rad 4% to 15% Mini-PROTEAN precast polyacrylamide gel. The gels were run at 100V constant voltage for 1 h and 15 min in running buffer (3 g Tris base, 14.4 g glycine, and 1 g SDS in 1l ionised water). Afterwards, proteins were transferred to a FL-PVDF membrane using the Towbin transfer buffer (25 mM Tris, 192 mM glycine, 20% methanol (v/v) (pH 8.3)). Membrane transfer was performed at room temperature, constant voltage 100V for 1 h and 15 min. Membranes were blocked in 5% milk for 1 h at room temperature. Membranes were washed twice TBST (50 mM Tris HCl, 150 mM NaCl, and 0.5% (w/v) Tween 20) and incubated 1 h and 30 min with the primary antibody anti-human CD49b, anti-p38 MAPK, anti-phospho-p38 MAPK, and mouse anti-human GAPDH in TBST. Membranes were washed thrice in TBST (10 min/wash). Secondary antibodies were then applied for 1 h at room temperature. Anti-mouse IgG secondary LiCOR IR Dye 800 antibody was diluted 1:30,000 in 0.01% (w/v) SDS TBST. Next, 3 TBST washes were performed and the last wash with deionized water. A LiCOR Odyssey Sa system was used for imaging the membranes. The intensity of the bands was quantified with Image Studio Lite software. Bands were normalised to GAPDH intensity.

## Cell migration

For p38 inhibitor experiments, cells were serum starved for 16 to 18 h. Cells were detached using TrypLE and neutralised with serum-free media, and $5 \times 10^4$ MDA-MB-231 cells per well were seeded into a CDM-coated 12-well plate. Migration experiments were carried out with 5% FBS DMEM. DMSO, p38 MAPK inhibitors (10 µM and 50 µM SB202190 and SB203580), and PP2A inhibitor (50 nM okadaic acid) were added at the time of cell seeding. Time-lapse imaging was started after a 6-h incubation. Bafilomycin A1 and E64d were added at the time of seeding in 10% FBS DMEM. For knockdown experiments, cells were plated in complete media and allowed to adhere for 6 h before imaging. Plates were imaged in a Nikon Inverted Ti eclipse with Oko-lab environmental control chamber with a 10×/NA 0.45 objective. Cells were incubated at 37°C and 5% $CO_2$; images were acquired every 10 min for at least 7 h and more than 40 cells per well were quantified per biological replicate. Individual cell migration was manually tracked using MTrack2, a Fiji/ImageJ plugin. The chemotaxis tool plugin in Fiji/Image J (https://ibidi.com/chemotaxis-analysis/171-chemotaxis-and-migration-tool.html) was used to calculate the velocity and directionality of migrating cells.

## Wound healing

For wound healing experiments, SW1990 cells were detached using Trypsin-EDTA and neutralised using 10% FBS DMEM. Cells were seeded into a 12-well plate and incubated overnight to achieve confluency. Monolayers were manually scratched into a cross shape with the help of a 200 µl pipette tip. Cells were covered by 0.5 mg/ml collagen I in 10% FBS DMEM (300 µl/

well) and incubated for 30 min at 37˚C to polymerise. After this time, 1 ml media was added on top of the scratches in the presence of the vehicle (DMSO), 10 μM BTT-3033 or 50 μM SB203580. Each branch of the scratches was imaged at 0 h, 3 h, and 6 h after adding the inhibitors using an Olympus Inverted Fluorescence Microscope (4× objective). The area of the scratch was calculated by tracing the perimeter of the non-invaded area using Fiji/Image J. The invasion ratio was normalised to the area at time 0 h.

## mRNA expression (RT-qPCR)

mRNA was extracted from snap-frozen samples, and stored at −80˚C, according to the manufacturer's protocol (RNeasy Mini–Qiagen). For one-way qPCR (analysis of MAP3K1, Hs_MAP3K1_1_SG QuantiTect Primer Assay, Qiagen), Luna Universal One-Step RT-qPCR Kit was used. Samples were prepared by adding less than 1 μg RNA, 5 μl Luna Universal One-Step Reaction Mix, 0.5 μl Luna WarmStart RT Enzyme Mix, 0.8 mix forward and reverse primer and toped up with RNase-free water for a total of 10 μl. For two-way qPCR (analysis of MAPK11; Hs_MAPK11_1_SG QuantiTect Primer Assay, Qiagen), cDNA was first synthesised with High-Capacity cDNA Reverse Transcription Kit (Fisher). Afterwards, loading master mix containing 5 μl QuantiNova SYBR Green PCR Kit (Qiagen) master mix, 1 μl forward and reverse primer, and 1 μl RNase free water was prepped and mixed with 3 μl cDNA solution (5 ng/μl). Finally, for both methodologies used, 10 μl of the sample was loaded into a 384-well plate. Quantstudio 12K flex real-time PCR system was used in the SYBR mode to analyse the samples. Expression levels were calculated using the $2^{−ΔΔC_T}$ method [114]. GAPDH was used as a control (Hs_Gapdh_3_SG QuantiTect Primer Assay, Qiagen). Each sample was tested in 3 technical replicates.

## DNA transfection

MDA-MB-231 stably expressing GFP was generated as described in [5]. Briefly, $8 \times 10^5$ cells/ well were seeded into a 6-well plate in 2 ml of 10% FBS DMEM without antibiotics. Confluent cells were transfected with 2.5 μg of pSBtet-GB GFP Luciferase plasmid and 0.25 μg of the sleeping beauty transposon plasmid, pCMV(CAT)T7-SB100, and 250 μl of OptiMEM, 5 μl p3000, and 3.75 μl Lipofectamine 3000, together with both plasmids were added on top of the 2 ml. Media was changed after 6 h. After 48 h, cells were selected with 2 μg/ml blasticidin. After selection, cells were FACS sorted.

## 3D ECM internalisation and 3D spheroids

3D spheroids were generated by the hanging drop method, previously described in [115], and 2,000 cells per 20 μl drop containing 4.8 mg/ml methylcellulose (Sigma-Aldrich) and 20 μg/ml soluble collagen I (BioEngineering) were pipetted on the lid of tissue culture dishes. Lids were turned and put on top of the bottom reservoir of the dish, which was filled with PBS to prevent evaporation. After 48 h, spheroids were embedded in 40 μl of 3 mg/ml rat tail collagen I (Corning) and 3 mg/ml Matrigel (Corning). For 3D uptake assays, ⅕ (v/v) of the matrix solution was labelled with a final concentration 20 μg/ml pHrodo containing 0.1 M sodium bicarbonate. Cells were imaged live every 24 h until day 3 post-embedding for α2 integrin knockdown and BTT-3033 treatment. For MAP3K1 knockdown, spheroids were imaged until day 4. Nikon A1 confocal (10× objective) was used to image whole spheroids. For invasion analysis, spheroids were thresholded, and the area of the selected threshold was quantified in Fiji/Image J. The invasion rate was defined as the ratio between the invasion area and the total area. For 3D uptake, the pHrodo-ECM intensity in the spheroid was quantified.

## Overall survival, relapse-free survival, and ROC analysis

The survival analysis was performed using Kaplan–Meier plotter (https://kmplot.com/analysis/), which can assess the effect of genes of interest on survival in 21 cancer types, including pancreatic cancer. Sources for the databases include Gene Expression Omnibus (GEO), European Genome-Phenome Archive (EGA), and The Cancer Genome Atlas (TCGA) [116]. RNA sequencing data from pancreatic ductal adenocarcinoma and normal pancreas was performed using TNMplot (tnmplot.com), which enables a direct comparison of tumour and normal samples and runs a Mann–Whitney U test [117]. The ROC plotter was used to analyse the link between gene expression and response to chemotherapy (including, taxane, anthracycline, ixabepilone, CMF, FAC, and FEC) using transcriptome-level data of breast cancer patients (https://www.rocplot.com/) [57]. Breast cancer data sets were identified in GEO (https://www.ncbi.nlm.nih.gov/gds), using the platform IDs "GPL96," "GPL570," and "GPL571" and the keywords "breast," "cancer," and "therapy." For genes with multiple probes, Jetset was used to select the most reliable probe set (https://services.healthtech.dtu.dk/services/jetset/).

## Immunofluorescence of FFPE tissue samples

Samples were provided by SEARCHBreast (https://searchbreast.org/). Tissue slides were deparaffinized and rehydrated. After antigen retrieval with sodium citrate buffer at pH 6, the slides were blocked for endogenous peroxidase with 3% hydrogen peroxide, permeabilised with 1% FBS, 0.5% Triton X-100 in PBS and blocked for non-specific binding with 5% FBS, 0.5% Triton X-100 in PBS. Staining for α2 integrin (Alexa-Fluor 488 anti-mouse CD49b, 1:200 in 5% FBS, 3% bovine serum albumin, 0.5% Triton X-100 in PBS) was performed overnight. Alternatively, slides were stained for collagen I (1:200 in 1% FBS PBST) overnight and with Alexa-Fluor 488 anti-rabbit IgG (1:500 in 1% FBS PBST) for 1 h. Slides were counterstained with Hoechst-33342 and then mounted (ProLong Gold Antifade Mountant) before fluorescence imaging with Nikon Inverted Ti eclipse with a 10×/NA 0.45 objective. The mean intensity of the mammary glands or tumour cells from the tissue sections was quantified using Fiji/ImageJ.

## Statistical analysis

All the gathered data was normalised to the control population. Data representation and statistical analysis were performed in GraphPad Prism (Version 9.4.1) software. Scatter plot data are represented by SuperPlots, which enables the incorporation of cell-level data and experimental repeatability in a single diagram [118]. Cell-level data is represented by dots. For images acquired at Nikon A1 confocal, cell data is colour-coded in blue, red, and cantaloupe to display individual biological replicates. SuperPlots include the average (mean) data in each biological replicate for Nikon A1 confocal data. Average values or well data is represented by squares. For images acquired in Opera Phenix microscope, blue, red, and cantaloupe squares show mean biological replicates. Multiple squares with the same colour represent technical replicates for each well. To compare 2 data sets, an unpaired *t* test was used; to compare more than 2 data sets, nonparametric one-way ANOVA was performed. High-throughput screening data was normalised between the NT5 and NT5 in the presence of Bafilomycin A1 (Normalised index = (Matrigel uptake index—mean NT5)/(Mean NT5-Mean Bafilomycin A1)).

## Supporting information

**S1 Fig. The ECM was delivered to and degraded in the lysosomes. (a)** NMuMG and PyMT#1 cells were seeded on NHS Fluorescein-labelled 1 mg/ml matrigel for 12 h in the

presence or absence of 20 μM E64d, fixed, stained for actin and nuclei and imaged with a Nikon A1 confocal microscope. Scale bar, 20 μM. Matrigel uptake index was calculated with Image J. Data are presented as the normalised mean ± SD; $N$ = 3 independent experiments. ****$p < 0.0001$; Kruskal–Wallis test. (b) NMuMG and PyMT#1 cells were seeded on biotinylated CDM for 12 h in the presence or absence of 20 μm E64d, fixed, stained with streptavidin Alexa Fluor 488, Phalloidin Alexa Fluor 555 and DAPI, imaged and quantified as in (a). Scale bar, 20 μm. Data are presented as the normalised mean ± SD; $N$ = 3 independent experiments. ****$p < 0.0001$; Kruskal–Wallis test. (c, d) MDA-MB-231 cells were seeded on NHS Alexa Fluor 555-labelled 1 mg/ml matrigel for 3 h, 5 h, 8 h, and 12 h, fixed, stained for EEA1 (c) or LAMP2 (d), actin and nuclei and imaged as in (a). Scale bar, 10 μm. Colocalisation was quantified with Image J. Data are presented as the mean ± SEM; $N$ = 3 independent experiments. **$p = 0.0057$, ****$p < 0.0001$ (c); *$p = 0.0207$, **$p = 0.0069$, ***$p = 0.0003$ (d); One-way ANOVA/Tukey's multiple comparisons test. (e) MDA-MB-231 cells were seeded on NHS-fluorescein (green) and pHrodo-labelled (magenta) 1 mg/ml collagen I and imaged live for 5 h. Representative time frames from S1 Video are shown. Scale bar, 5 μm. All the raw data associated with this figure are available in S8 Data.
(TIF)

**S2 Fig. A kinome and phosphatome screen identified regulators of matrigel internalisation.** (a) Normalised cloud plot analysis from replicate 2. (b) First derivative of the curve for hit threshold for replicate 2. (c) Correlation between replicate 1 and 2. (d) Representative images of matrigel uptake in cells transfected with a non-targeting siRNA control (siNT) and an siRNA targeting PAK1 (siPAK1). Scale bar, 20 μm. (e) Data are presented as the normalised mean ± SD; $N$ = 6 replicates from 2 independent experiments. ****$p < 0.0001$; Kruskal–Wallis test. (f) Screening plates were fixed and stained for β1 integrin (green) and nuclei (blue) and imaged with 40× Opera Phenix microscope; scale bar, 60 μm. Signal intensity was quantified with Columbus Software. *$p = 0.0286$; Mann–Whitney test. (g) MDA-MB-231 cells were transfected with an siRNA targeting PAK1 (siPAK1) or a non-targeting siRNA control (siNT). PAK1 and GAPDH protein levels was quantified by western blotting. Data are presented as the normalised mean ± SD; $N$ = 4 independent experiments; *$p = 0.0286$; Mann–Whitney test. (h) Z' robust and standard for screening validation. (i) Diagram of MAPK activation pathways. (j) Heatmap of major MAPKs in the kinome and phosphatome screen. $N$ = 2 biological replicates. Representative images for MAPK11. Scale bar, 20 μm. All the raw data associated with this figure are available in S9 Data.
(TIF)

**S3 Fig. PP2A regulates collagen I uptake and cell morphology.** (a) Representative scheme of Protein Phosphatase 2A (PP2A) subunits. (b) Heatmap of major PP2A subunits in the kinome and phosphatome screen. $N$ = 2 biological replicates. (c) $3 \times 10^5$ MDA-MB-231 cells were cultured on pHrodo-labelled collagen I for 6 h in the presence of 50 nM okadaic acid (Okad Ac.) or water (Control), stained with 1 μg/ml Hoechst and imaged live. Scale bar, 20 μm. Collagen I uptake index was measured with Image J. Values represented are normalised mean + SD from $N$ = 3 independent experiments; ****$p < 0.0001$; Kruskal–Wallis test. (d–g) MDA-MB-231 (d) and A2780-Rab25 (e) cells were transfected with an siRNA targeting PPP2R1A (siPPP2R1A) or a non-targeting siRNA control (siNT) and seeded on CDM. MDA-MB-231 (f) and A2780-Rab25 (g) were seeded on CDM in the presence of 50 nM Okadaic acid or the control (water). Cells were imaged live with a 10× Nikon Inverted Ti eclipse with Oko-lab environmental control chamber for 17 h. Aspect ratio (AR) was calculated with Image J at 6 h, 12 h, and 18 h after cell seeding. Scale bar, 50 μm. Values represented are AR from single cells, mean AR + SD from $N$ = 3 independent experiments; ****$p < 0.0001$; Kruskal–Wallis test. All the

raw data associated with this figure are available in S10 Data.
(TIF)

**S4 Fig. α2 and β1 integrin mediated ECM internalisation in cancer cells. (a)** MDA-MB-231 cells transfected with an siRNA targeting β1 integrin (siITGB1) or a non-targeting siRNA control (siNT), plated on pHrodo-labelled 0.5 mg/ml matrigel for 6 h, stained with 1 μg/ml Hoechst and imaged live. Data are presented as the normalised mean ± SD; $N = 3$ independent experiments. ****$p < 0.0001$; Mann–Whitney test. **(b)** MDA-MB-231 cells were seeded on NHS Alexa Fluor 555-labelled 1 mg/ml matrigel for 3 h, 5 h, 8 h, and 12 h, fixed and stained for β1 integrin (ITGB1), actin and nuclei. Scale bar, 10 μm. Data are presented as the mean ± SEM; $N = 3$ independent experiments. **(c, d)** A2780-Rab25 cells were transfected with an siRNA targeting α2 integrin (siITGA2) or a non-targeting siRNA control (siNT), seeded on pHrodo-labelled CDM (c) or collagen I (d) for 6 h, stained with 1 μg/ml Hoechst and imaged live. Data are presented as the normalised mean ± SD; $N = 3$ independent experiments. ****$p < 0.0001$; Mann–Whitney test. **(e)** YEJ P cells were allowed to adhere to pHrodo-labelled 1 mg/ml collagen I for 2 h, treated with 5 μM BTT-3033 or DMSO for 14 h, stained with 1 μg/ml Hoechst and imaged live. Data are presented as the normalised mean ± SD; $N = 3$ independent experiments. ****$p < 0.0001$; Mann–Whitney test. **(f)** SW1990 cells were seeded on pHrodo-labelled 1 mg/ml collagen I for 2 h, treated with 10 μM BTT-3033 or DMSO for 4 h, stained with 1 μg/ml Hoechst and imaged live. Data are presented as the normalised mean ± SD; $N = 3$ independent experiments. ****$p < 0.0001$; Mann–Whitney test. **(g)** Cells were transfected with an siRNA targeting α2 integrin (siITGA2) or a non-targeting siRNA control (siNT) for 72 h, lysed and α2 integrin and GAPDH protein levels were measured by western blotting. Data are presented as the normalised mean ± SD; $N = 4$ independent replicates. *$p = 0.0286$; Mann–Whitney test. All the raw data associated with this figure are available in S11 Data.
(TIF)

**S5 Fig. p38 MAPK inhibition reduced ECM internalisation. (a)** Schematic representation of MAPK inhibitors. **(b)** MDA-MB-231 cells were serum starved for 16 to 18 h, and $10^4$ cells were seeded on pHrodo-labelled ECM for 6 h in the presence of DMSO or MAPK inhibitors in 0% FBS, stained with 1 μg/ml Hoechst and imaged live. Data analysis was performed with Columbus software. **(c)** A2780-Rab25 cells were serum starved for 16 to 18 h; $3 \times 10^5$ cells were seeded on pHrodo-labelled CDM for 6 h in the presence of DMSO, 10 μM or 50 μM SB202190 in 5% FBS, stained with Hoechst and imaged live. Scale bar, 20 μm. CDM uptake was quantified with ImageJ. Values represented are normalised mean + SD from $N = 3$ independent experiments; ****$p < 0.0001$; Kruskal–Wallis test. **(d)** A2780-Rab25 cells were serum starved for 16 to 18 h, and $3 \times 10^5$ cells were cultured on 1 mg/ml collagen I, labelled with NHS-Alexa fluor 555, for 6 h in the presence of DMSO, 10 μM or 50 μM SB202190 in 5% FBS. Cells were fixed and stained for actin and nuclei. Scale bar, 20 μm. Collagen I uptake was quantified with ImageJ. Values represented are normalised mean + SD from $N = 3$ independent experiments; ****$p < 0.0001$; Kruskal–Wallis test. **(e)** YEJ P cells were serum starved for 18 h, and $3 \times 10^5$ cells were seeded on pHrodo-labelled 1 mg/ml collagen I for 16 h in the presence of DMSO or 10 μM SB203580 in 5% FBS, stained with Hoechst and imaged live. Scale bar, 20 μm. Collagen I uptake was quantified with ImageJ. Values represented are normalised mean + SD from $N = 3$ independent experiments; ****$p < 0.0001$; Mann–Whitney test. **(f)** SW1990 cells were serum starved for 18 h, and $3 \times 10^5$ cells were cultured on pHrodo-labelled 1 mg/ml collagen I for 6 h in the presence of DMSO, 10 μM or 50 μM SB203580 in 5% FBS, stained with Hoechst and imaged live. Scale bar, 20 μm. Collagen I uptake was quantified with ImageJ. Values represented are normalised mean + SD from $N = 3$ independent experiments;

****$p < 0.0001$; Kruskal–Wallis test. All the raw data associated with this figure are available in S12 Data.
(TIF)

**S6 Fig. MAP3K1, MAPK11, and PPP2R1A knockdown decreased ECM internalisation. (a)** A2780-Rab25 cells were transfected with an siRNA targeting MAPK3K1 (siMAP3K1), an siRNA targeting MAPK11 (siMAPK11), an siRNA targeting PPP2R1A (siPPP2R1A), or a non-targeting siRNA control (siNT), seeded on pHrodo-labelled CDM for 6 h, stained with 1 µg/ml Hoechst and imaged live. Scale bar, 20 µm. **(b)** CDM uptake index was calculated with Image J. Values represented are normalised mean + SD from $N = 3$ independent experiments; ****$p < 0.0001$; Kruskal–Wallis test. **(c)** A2780-Rab25 cells were transfected as in (a), seeded on 1 mg/ml collagen I, labelled with NHS-Alexa fluor 555, for 6 h, fixed and stained for nuclei. Scale bar, 20 µm. **(d)** Collagen I uptake index was calculated with Image J. Values represented are normalised mean + SD from $N = 3$ independent experiments; ****$p < 0.0001$; Kruskal–Wallis test. **(e, f)** MDA-MB-231 (e) and A2780-Rab25 (f) cells were transfected with an siRNA targeting MAP3K1 (siMAP3K1) or a non-targeting siRNA control (siNT), RNA was extracted and MAP3K1 expression was quantified by qPCR. Normalised data from $N = 3$ independent experiments; **$p = 0.0022$; ***$p = 0.0003$; Mann–Whitney test. **(g, h)** MDA-MB-231 (g) and A2780-Rab25 (h) cells were transfected with an siRNA targeting MAPK11 (siMAPK11) or a non-targeting siRNA control (siNT), RNA was extracted and MAPK11 expression was quantified by qPCR. Normalised data from $N = 3$ independent experiments; ****$p < 0.0001$; Kruskal–Wallis test. **(i)** MDA-MB-231 cells overexpressing PPP2R1A-GFP were transfected with an siRNA targeting PPP2R1A (siPPP2R1A) or a non-targeting siRNA control (siNT), fixed and stained for nuclei. GFP normalised mean intensity + SD from $N = 3$ independent experiments is shown; ****$p < 0.0001$; Mann–Whitney test. **(j, k)** MDA-MB-231 cells were transfected with an siRNA targeting MAPK11 (siMAPK11, j), an siRNA targeting PPP2R1A (siPPP2R1A, k) or a non-targeting siRNA control (siNT), α2 integrin (ITGA2) and GAPDH protein levels was quantified by western blotting. Data are presented as the normalised mean ± SD; $N = 4$ independent experiments; *$p = 0.0286$; Mann–Whitney test. All the raw data associated with this figure are available in S13 Data.
(TIF)

**S7 Fig. Individual MAPK11 siRNAs impaired collagen I uptake. (a)** MDA-MB-231 cells were transfected with 3 individual siRNA targeting MAPK11 (siMAPK11-7, 9, and 10) or a non-targeting siRNA control (siNT), seeded on pHrodo-labelled 1 mg/ml collagen I for 6 h, stained with 1 µg/ml Hoechst and imaged live. Scale bar, 20 µm. **(b)** Collagen I uptake index was calculated with Image J. Values represented are normalised mean + SD from $N = 3$ independent experiments; ****$p < 0.0001$; Kruskal–Wallis test. **(c)** MDA-MB-231 cells were transfected as in (a), RNA was extracted and MAPK11 expression was quantified by qPCR. Normalised data from $N = 4$ independent experiments; ****$p < 0.0001$; **$p = 0.0065$; Kruskal–Wallis test. All the raw data associated with this figure are available in S14 Data.
(TIF)

**S8 Fig. Collagen I-bound α2β1 integrin was degraded in the lysosomes following internalisation. (a)** MDA-MB-231 cells were seeded on NHS-Alexa Fluor 555-labelled 1 mg/ml collagen I for 24 h in the presence of 20 mM E64d, 200 nM Bafilomycin A1 (BafA1) or DMSO control, stained with 1 µg/ml Hoechst and imaged live. Scale bar, 20 µm. **(b)** Collagen I internal pool was calculated with Image J. Values represented are normalised mean + SD from $N = 3$ independent experiments; ****$p < 0.0001$; Kruskal–Wallis test. **(c)** MDA-MB-231 cells treated as in (a), fixed and stained for α2 integrin (ITGA2, green), actin (red), and nuclei

(blue). Scale bar, 20 μm. **(d)** α2 integrin internal pool was calculated with Image J. Values represented are normalised mean + SD from $N = 3$ independent experiments; ****$p < 0.0001$; Kruskal–Wallis test. **(e)** MDA-MB-231 cells were seeded on 1 mg/ml collagen I for 24 h in the presence of 200 nM Bafilomycin A1 (BafA1) or DMSO control, α2 integrin (ITGA2), and GAPDH protein levels was quantified by western blotting. Data are presented as the normalised mean ± SD; $N = 4$ independent experiments; *$p = 0.0286$; Mann–Whitney test. All the raw data associated with this figure are available in S15 Data.
(TIF)

**S9 Fig. p38 and a2 integrin inhibition reduced macropinocytosis of soluble dextran in MDA-MB-231 cells. (a)** Schematic representation of the experimental set up. **(b)** MDA-MB-231 cells were seeded on 0.1 mg/ml collagen I for 6 h, pretreated with DMSO (vehicle) or 50 μm SB202190 for 30 min, incubated with 0.25 mg/ml rhodamine-dextran (red) for 1 h in the presence of DMSO or SB202190, fixed and stained for β1 integrin (ITGB1) and nuclei. Scale bar, 20 μM. Dextran uptake index was measured with Image J. Data are presented as the normalised mean ± SD; $N = 3$ independent experiments. ****$p < 0.0001$; Mann–Whitney test. **(c)** MDA-MB-231 cells were seeded on 0.1 mg/ml collagen I for 6 h in the presence of DMSO (vehicle) or 10 μM BTT-3033, incubated with 0.25 mg/ml rhodamine-dextran (red) for 1 h, fixed and stained for β1 integrin (ITGB1) and nuclei. Scale bar, 20 μm. Dextran uptake index was measured with Image J. Data are presented as the normalised mean ± SD; $N = 3$ independent experiments. ****$p < 0.0001$; Mann–Whitney test. **(d)** MDA-MB-231 cells were transfected with an siRNA targeting α2 integrin (siITGA2) or a non-targeting siRNA control (siNT), seeded on 0.1 mg/ml collagen I for 6 h, incubated with 0.25 mg/ml rhodamine-dextran (red) for 1 h, fixed and stained for β1 integrin (ITGB1) and nuclei. Scale bar, 20 μm. Data are presented as the normalised mean ± SD; $N = 3$ independent experiments. ****$p < 0.0001$; Mann–Whitney test. All the raw data associated with this figure are available in S16 Data.
(TIF)

**S10 Fig. Regulators of ECM internalisation were required for ovarian and pancreatic cancer cell migration. (a–c)** A2780-Rab25 cells were transfected with an siRNA targeting MAP3K1 (siMAP3K1), an siRNA targeting MAPK11 (siMAPK11), an siRNA targeting PPP2R1A (siPPP2R1A), an siRNA targeting α2 integrin (si-ITGA2), or a non-targeting siRNA control (siNT), seeded on CDM for 6 h and imaged live with a 10× Nikon Inverted Ti eclipse with Oko-lab environmental control chamber for 17 h. Spider plots show the migration paths of manually tracked cells (directionality >0.5 in black, <0.5 in red). Box and whisker plots represent 5–95 percentile, + represents the mean, dots are <5% and >95%; $N = 3$ independent experiments. ****$p < 0.0001$; Kruskal–Wallis test. **(d)** A2780-Rab25 cells were seeded on CDM for 6 h in the presence of DMSO (Ctrl.) or 50 μM SB203580 and imaged live with a 10× Nikon Inverted Ti eclipse with Oko-lab environmental control chamber for 17 h. Spider plots show the migration paths of manually tracked cells (directionality >0.5 in black, <0.5 in red). Box and whisker plots represent 5–95 percentile, + represents the mean, dots are <5% and >95%; $N = 3$ independent experiments. ****$p < 0.0001$; Kruskal–Wallis test. **(e)** A2780-Rab25 cells were seeded on CDM for 6 h in the presence of the vehicle (water, Ctrl.) and 50 nM Okadaic acid (Okad Ac.) and imaged live with a 10× Nikon Inverted Ti eclipse with Oko-lab environmental control chamber for 17 h. Spider plots show the migration paths of manually tracked cells (directionality >0.5 in black, <0.5 in red). Box and whisker plots represent 5–95 percentile, + represents the mean, dots are <5% and >95%; $N = 3$ independent experiments. ****$p < 0.0001$; Kruskal–Wallis test. **(f)** SW1990 cell confluent monolayers were scratched and overlaid with 0.5 mg/ml collagen I and cells were imaged at 0 h, 3 h, and 6 h. Yellow lines indicate the wound edges. The bar graph shows the normalised relative gap area ± SEM. $N = 4$

independent experiments. *$p \leq 0.0213$, **$p = 0.0025$; 2-way ANOVA. All the raw data associated with this figure are available in S17 Data.
(TIF)

**S11 Fig. ECM lysosomal degradation was required for breast cancer cell migration.**
MDA-MB-231 cells were seeded on CDM in the presence of 200 nM Bafilomycin A1 (BafA1), 20 μM E64d, or DMSO control for 6 h and imaged live with a 10× Nikon Inverted Ti eclipse with Oko-lab environmental control chamber for 17 h. Spider plots show the migration paths of manually tracked cells (directionality >0.5 in black, <0.5 in red). Bar, 200 mm. Box and whisker plots represent 5–95 percentile, + represents the mean, dots are <5% and >95%; $N = 1$ independent experiments. ***$p = 0.0002$, ****$p < 0.0001$; Kruskal–Wallis test. All the raw data associated with this figure are available in S18 Data.
(TIF)

**S12 Fig. NHE1 was required for collagen I uptake and directional cell migration. (a)** Schematic, NHE1 promotes macropinocytosis. **(b)** MDA-MB-231 cells were transfected with an siRNA targeting NHE1 (siNHE1) or a non-targeting siRNA control (siNT), seeded on pHrodo-labelled 1 mg/ml collagen I for 6 h, stained with 1 μg/ml Hoechst and imaged live. Scale bar, 30 μm. Collagen I uptake index was calculated with Image J. Values represented are normalised mean + SD from $N = 3$ independent experiments; ****$p < 0.0001$; Mann–Whitney test. **(c)** MDA-MB-231 cells were transfected as in (a), seeded on CDM and imaged live with a 10× Nikon Inverted Ti eclipse with Oko-lab environmental control chamber for 17 h. Spider plots show the migration paths of manually tracked cells (directionality >0.5 in black, <0.5 in red). Bar, 200 mm. Box and whisker plots represent 5–95 percentile, + represents the mean, dots are <5% and >95%; $N = 3$ independent experiments. ****$p < 0.0001$; Kruskal–Wallis test. **(d)** Cells were transfected as in (a) for 72 h, lysed and NHE1 and GAPDH protein levels were measured by western blotting. Data are presented as the normalised mean ± SD; $N = 4$ independent replicates. *$p = 0.0286$; Mann–Whitney test. All the raw data associated with this figure are available in S19 Data.
(TIF)

**S13 Fig. α2 integrin is up-regulated in mouse mammary tumours and MAP3K1, MAPK11, PPP2R1A, and α2 integrin expression is higher in chemotherapy-resistant breast cancer patients. (a, b)** Tissue sections from polyoma middle T-derived mouse mammary tumours were stained for collagen I (black) and nuclei (blue) (a) or α2 integrin (ITGA2, b); * highlights the stroma, arrow-head indicate collagen I-positive vesicles. Scale bar, 10 μm (a) and 50 μm (b). **(c)** α2 integrin mean intensity was quantified with Image J; $N = 3$ independent experiments. ****$p < 0.0001$; Kruskal–Wallis test. **(d–g)** RNA sequencing data and ROC analysis for MAP3K1 (a), MAPK11 (b), PPP2R1A (c), and α2 integrin (ITGA2, d) from chemoresistant (non-responder (NR)) and chemosensitive (responder (R)) breast cancer tumours. **$p = 0.0090$; ***$p = 0.0002$; ****$p < 0.0001$; Mann–Whitney test. All the raw data associated with this figure are available in S20 Data.
(TIF)

**S1 Video. MDA-MB-231 cells internalised and acidified collagen I.** MDA-MB-231 cells were seeded on NHS-fluorescein (green) and pHrodo-labelled 1 mg/ml collagen I (magenta), cells were allowed to adhere for 30 min before being imaged live for 5 h every minute with a 63× oil objective Zeiss LSM980 Airyscan 2 microscope with environmental chamber. Video corresponding to S1E Fig.
(AVI)

**S2 Video. MDA-MB-231 cells internalised collagen I during cell migration.** MDA-MB-231 cells were seeded on NHS-fluorescein (green) and pHrodo-labelled 1 mg/ml collagen I (magenta), cells were allowed to adhere for 30 min and imaged live for 5 h every minute with a 63× oil objective Zeiss LSM980 Airyscan 2 microscope with environmental chamber. Video corresponding to Fig 5a.
(MOV)

**S3 Video. MDA-MB-231 cells internalised CDM during cell migration.** MDA-MB-231 cells were seeded on pHrodo-labelled CDM (red) for 6 h before imaging for an additional period of 6 h with a 10× objective from a Nikon Inverted Ti eclipse with Oko-lab environmental control chamber. Video corresponding to Fig 5b.
(AVI)

**S1 Table. Raw data values for Matrigel uptake index.** Data were normalised between NT5 +BafA1 (-1) and NT5 (0). Hits are shown in blue, while controls are in green.
(XLSX)

**S2 Table. Reactome analysis of the positive and negative regulator hits obtained in the screening.**
(XLSX)

**S3 Table. Data were normalised between NT5+BafA1 (-1) and NT5 (0).** The table shows kinome and phosphatome results and deconvolution of 4 individual siRNAs (2 biological replicates, 2 technical replicates per biological replicate). Validated positive regulators are highlighted in orange, while negative regulators are in green. Controls (ITGB1 and PAK1) are in dark red, NT5+BafA1 is in pink, and NT5 is blue.
(XLSX)

**S4 Table. Cell seeding values for generation of cell derived matrices.**
(XLSX)

**S5 Table. siRNA information.**
(XLSX)

**S1 Raw Images. Original western blot images used to prepare Figs 3f, S2g, S4g, S6j, S6k, S8e, and S12d.**
(PDF)

**S1 Data. Numerical data used for the generation of the graphs presented in Fig 1.** The biological replicates are colour coded.
(XLSX)

**S2 Data. Numerical data used for the generation of the graphs presented in Fig 2.** The biological replicates are colour coded.
(XLSX)

**S3 Data. Numerical data used for the generation of the graphs presented in Fig 3.** The biological replicates are colour coded.
(XLSX)

**S4 Data. Numerical data used for the generation of the graphs presented in Fig 4.** The biological replicates are colour coded.
(XLSX)

**S5 Data. Numerical data used for the generation of the graphs presented in Fig 5.** The biological replicates are colour coded.
(XLSX)

**S6 Data. Numerical data used for the generation of the graphs presented in Fig 6.** The biological replicates are colour coded.
(XLSX)

**S7 Data. Numerical data used for the generation of the graphs presented in Fig 7.** The biological replicates are colour coded.
(XLSX)

**S8 Data. Numerical data used for the generation of the graphs presented in S1 Fig.** The biological replicates are colour coded.
(XLSX)

**S9 Data. Numerical data used for the generation of the graphs presented in S2 Fig.** The biological replicates are colour coded.
(XLSX)

**S10 Data. Numerical data used for the generation of the graphs presented in S3 Fig.** The biological replicates are colour coded.
(XLSX)

**S11 Data. Numerical data used for the generation of the graphs presented in S4 Fig.** The biological replicates are colour coded.
(XLSX)

**S12 Data. Numerical data used for the generation of the graphs presented in S5 Fig.** The biological replicates are colour coded.
(XLSX)

**S13 Data. Numerical data used for the generation of the graphs presented in S6 Fig.** The biological replicates are colour coded.
(XLSX)

**S14 Data. Numerical data used for the generation of the graphs presented in S7 Fig.** The biological replicates are colour coded.
(XLSX)

**S15 Data. Numerical data used for the generation of the graphs presented in S8 Fig.** The biological replicates are colour coded.
(XLSX)

**S16 Data. Numerical data used for the generation of the graphs presented in S9 Fig.** The biological replicates are colour coded.
(XLSX)

**S17 Data. Numerical data used for the generation of the graphs presented in S10 Fig.** The biological replicates are colour coded.
(XLSX)

**S18 Data. Numerical data used for the generation of the graphs presented in S11 Fig.** The biological replicates are colour coded.
(XLSX)

**S19 Data. Numerical data used for the generation of the graphs presented in S12 Fig.** The biological replicates are colour coded.
(XLSX)

**S20 Data. Numerical data used for the generation of the graphs presented in S13 Fig.** The biological replicates are colour coded.
(XLSX)

## Acknowledgments

Imaging work was performed at the Wolfson Light Microscopy Facility, University of Sheffield, using the Nikon A1 confocal, Nikon widefield and Airyscan microscope. High-throughput imaging was performed in the RNAi screening facility in IMCB, Singapore. FACS sorting was performed by SIgN (A*STAR, Singapore). qPCR analysis was performed in collaboration with the Tsakiridis lab at the University of Sheffield. We would like to acknowledge Marga Albu for developing the Image J macro used for ECM uptake quantification. We thank Dr. Rebecca Bennion for the critical reading and proofreading of the manuscript.

The Wolfson Light Microscopy Facility, University of Sheffield, is funded by the Wellcome Trust (grant WT093134AIA).

## Author Contributions

**Conceptualization:** Montserrat Llanses Martinez, Elena Rainero.

**Data curation:** Montserrat Llanses Martinez, Joe Tyler.

**Formal analysis:** Montserrat Llanses Martinez.

**Funding acquisition:** Frederic A. Bard, Elena Rainero.

**Investigation:** Montserrat Llanses Martinez, Keqian Nan, Zhe Bao, Rachele Bacchetti, Shengnan Yuan, Joe Tyler.

**Methodology:** Montserrat Llanses Martinez, Keqian Nan, Zhe Bao, Rachele Bacchetti, Joe Tyler, Xavier Le Guezennec.

**Project administration:** Elena Rainero.

**Supervision:** Xavier Le Guezennec, Frederic A. Bard, Elena Rainero.

**Writing – review & editing:** Xavier Le Guezennec, Elena Rainero.

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
