## [Editor Report · Decision Letter 0]

22 May 2024

Dear Dr Rainero, 

Thank you for submitting your manuscript entitled "Internalisation of integrin-bound extracellular matrix modulates invasive carcinoma cell migration" for consideration as a Research Article by PLOS Biology.

Your manuscript, along with the Review Commons' reports and your revision plan, has now been evaluated by the PLOS Biology editorial staff as well as by an academic editor with relevant expertise, and I am writing to let you know that we would like to invite you to submit a revision, but as a Methods and Resources paper. Thus, when you submit the metadata (see below), please select that type of article from the drop down menu.

Before we can send you the decision, we need you to complete your submission by providing the metadata that is required for full assessment. To this end, please login to Editorial Manager where you will find the paper in the 'Submissions Needing Revisions' folder on your homepage. Please click 'Revise Submission' from the Action Links and complete all additional questions in the submission questionnaire.

Once your full submission is complete, your paper will undergo a series of checks. After your manuscript has passed the checks, we will send you the decision. To provide the metadata for your submission, please Login to Editorial Manager (https://www.editorialmanager.com/pbiology) within two working days, i.e. by May 24 2024 11:59PM.

Kind regards,

Ines

--

Ines Alvarez-Garcia, PhD

Senior Editor

PLOS Biology

---

## [Editor Report · Decision Letter 1]

28 May 2024

Dear Dr Rainero,

Thank you for completing the metadata of the manuscript entitled "Internalisation of integrin-bound extracellular matrix modulates invasive carcinoma cell migration" and submitted via Review Commons as a Methods and Resources article in PLOS Biology.

Based on the associated reviews and the advice of an Academic Editor expert in the field, we would like to invite you to submit a revision that thoroughly address the reviewers' reports according to the revision plan.

Please note that we cannot make a decision about publication until we have seen the revised manuscript and that your revised manuscript is likely to be sent for further evaluation by all or a subset of the original reviewers.

**IMPORTANT - SUBMITTING YOUR REVISION**

3. Resubmission Checklist

a) *PLOS Data Policy*

b) *Published Peer Review*

d) *Blurb*

Please also provide a blurb which (if accepted) will be included in our weekly and monthly Electronic Table of Contents, sent out to readers of PLOS Biology, and may be used to promote your article in social media. The blurb should be about 30-40 words long and is subject to editorial changes. It should, without exaggeration, entice people to read your manuscript. It should not be redundant with the title and should not contain acronyms or abbreviations. For examples, view our author guidelines: https://journals.plos.org/plosbiology/s/revising-your-manuscript#loc-blurb

Sincerely,

Ines

--

Ines Alvarez-Garcia, PhD

Senior Editor

PLOS Biology

---

## [Decision Letter · Decision Letter 2]

1 Oct 2024

Dear Dr Rainero,

Thank you for your patience while we considered your revised manuscript entitled "Internalisation of integrin-bound extracellular matrix modulates invasive carcinoma cell migration" for publication as a Methods and Resources at PLOS Biology. This revised version of your manuscript has been evaluated by the PLOS Biology editors, the Academic Editor and the two original reviewers.

Based on the reviews (attached below), we are likely to accept this manuscript for publication, provided you satisfactorily address the data and other policy-related requests stated below.

In addition, we would like you to rework the Abstract keeping in mind the resource nature of the manuscript and we would also like you to consider a suggestiong to improve the title:

"Novel kinase regulators of extracellular matrix internalization identified by high-content screening modulate invasive carcinoma cell migration"

We expect to receive your revised manuscript within two weeks. 

*Published Peer Review History*

*Press*

Sincerely,

Ines

--

Ines Alvarez-Garcia, PhD

Senior Editor

PLOS Biology

Fig. 1B, C, E, F; Fig. 2A-D; Fig. 3B, D-F; Fig. 4B, D, E; Fig. 5C-F; Fig. 6C-G; Fig. 7A-I; Fig. S1A-D; Fig. S2A-C, E-H, J; Fig. S3B-G; Fig. S4A-H; Fig. S5B-F; Fig. S6B, D-K; Fig. S7B, C; Fig. S8B, D, E; Fig. S9B-D; Fig. S10A-F; Fig. S11; Fig. S12B-D and Fig. S13C-D

Please also ensure that figure legends in your manuscript include information ON WHERE THE UNDERLYING DATA CAN BE FOUND, and ensure your supplemental data file/s has a legend.

CODE POLICY

Reviewers' comments

Rev. 1: Johanna Ivaska - note that this reviewer has signed her review

The authors have addressed all of my concerns and I recommend publication of this interesting study.

Rev. 2:

The revised manuscript by Martinez et al. has addressed all of my previous concerns. I support publication. I commend the authors on a lovely manuscript.

There is one error to address. In the results for examination of ECM in tissue, this is referred to as Extended data Fig. 9. I believe that this should be Extended data Fig. 13.

---

## [Editor Report · Decision Letter 3]

6 Nov 2024

Dear Dr Rainero,

Thank you for the submission of your revised Methods and Resources entitled "Novel kinase regulators of extracellular matrix internalization identified by high-content screening modulate invasive carcinoma cell migration" for publication in PLOS Biology. On behalf of my colleagues and the Academic Editor, Carole Parent, I am delighted to let you know that we can in principle accept your manuscript for publication, provided you address any remaining formatting and reporting issues. These will be detailed in an email you should receive within 2-3 business days from our colleagues in the journal operations team; no action is required from you until then. Please note that we will not be able to formally accept your manuscript and schedule it for publication until you have completed any requested changes.

PRESS

Sincerely, 

Ines

--

Ines Alvarez-Garcia, PhD

Senior Editor

PLOS Biology
